# Transfer of learning: Analysis of dose-response functions from a large-scale, online, cognitive training dataset

**Allen M. Osman**[1], **Paul I. Jaffe**[1,2], **Nicole F. Ng**[1], **Kelsey R. Kerlan**[1], **Robert J. Schafer**[1] *

**1** Department of Research and Development, Lumos Labs, San Francisco, California, United States of America, **2** Department of Psychology, Stanford University, Stanford, California, United States of America

* bschafer@lumoslabs.com

**Data Availability Statement:** The dataset for this study is available on the Open Science Framework: https://doi.org/10.17605/OSF.IO/HDGMB.

## Abstract

Fundamental to the efficacy of cognitive training (CT) is its dose. Here we used the power and breadth afforded by a large dataset to measure precisely dose-response (D-R) functions for CT and to examine the generality of their magnitude and form. The present observational study involved 107,000 users of Lumosity, a commercial program comprising computer games designed to provide CT over the internet. In addition to training with Lumosity games, these users took an online battery of cognitive assessments (NeuroCognitive Performance Test, NCPT) on two or more occasions separated by at least 10 weeks. Changes in performance on the NCPT between the first and second assessments were examined as a function of the amount of intervening gameplay. The resulting D-R functions were obtained both for overall performance on the NCPT and performance on its eight subtests. Also examined were differences between D-R functions from demographic groups defined by age, gender, and education. Monotonically increasing D-R functions, well fit by an exponential approach to an asymptote, were found consistently for overall performance on the NCPT, performance on seven of the eight subtests, and at each level of age, education, and gender. By examining how individual parameters of the D-R functions varied across subtests and groups, it was possible to measure separately changes in the effects on NCPT performance of 1) transfer from CT and 2) direct practice due to repeated testing. The impact of both transfer and direct practice varied across subtests. In contrast, while the effects of direct practice diminished with age, those of transfer remained constant. Besides its implications for CT by older adults, this latter finding suggests that direct practice and transfer do not involve identical learning processes, with transfer being limited to learning processes that remain constant across the adult lifespan.

## Introduction

Cognitive training (CT), i.e., exercises that target specific cognitive functions or capacities, has engendered considerable interest and debate [1, 2]. Much of the interest stems from its

**Funding:** No external funding contributed to this research; Lumos Labs, Inc. funded the research through the development of its software tools and through the employment of AMO, PIJ, NFN, KRK, and RJS. The specific roles of these authors are articulated in the "Author Contributions" section. Other members of the company contributed suggestions and ideas during the design of the study and preparation of the manuscript. Lumos Labs had no other role in the study design, data collection and analysis, decision to publish, or preparation of the manuscript. Legal approval for publication before submission of the manuscript was obtained from Lumos Labs.

**Competing interests:** I have read the journal's policy and the authors of this manuscript have the following competing interests: The present study examined effects of cognitive training with Lumosity on the NeuroCognitive Performance Test, both of which are produced by Lumos Labs, Inc. At the time of the study, all authors were paid employees of the company, and all hold stock in the company. This does not alter our adherence to PLOS ONE policies on sharing data and materials.

possible use to enhance or preserve normal cognition or to ameliorate disorders of cognition associated with neuropsychological conditions or pharmacological treatments (e.g., [3–6]). Until recently, the primary question asked about CT was whether it "works" [7]. This question motivated research that focused on the presence, reliability, and interpretation of treatment effects in efficacy studies of CT. But increasing emphasis has begun to be placed on the related question of what factors influence the size and properties of the treatment effects observed in these studies [8]. Among such factors are ones concerning the CT program, trained population, and design features of the study. Besides shedding additional light on the properties and mechanisms (including possible placebo effects [9]) of CT, this approach promotes the development of more effective training and individualized treatment.

This shift in research emphasis has been supported by the availability of increasingly large datasets. Because most CT effects observed so far have been modest (Cohen's d < 0.5), discerning how they are modulated by a factor requires the sensitivity provided by large datasets. These large datasets have been made possible by two developments. First, the number of CT efficacy studies has increased over time, and the accumulating results of these studies have been combined in meta-analyses (e.g., [10–12]). Second, some individual studies include a large number of participants (e.g., [13–15]). The largest of these involve computerized CT conducted remotely over the internet [16–18]. Interestingly, many of these large studies and meta-analyses have been of cognitively healthy older adults (e.g., [10–14, 17, 19]). This makes sense in that ameliorating the normal cognitive decline associated with aging and preventing or delaying dementia are major concerns for this growing demographic [20–22].

The present study employed a large dataset to examine a factor fundamental to the transfer of learning from CT to other tasks: the amount of training or "dose." The power afforded by this dataset was used to measure precisely how the "response" on a cognitive assessment varied with dose of CT (i.e., dose-response or D-R functions). Analogous to learning curves, which provide a continuous measure of learning on a practiced task, D-R functions for CT provide a continuous measure of transfer from CT to other tasks. The form of these D-R functions (e.g., intercept, rate of increase, asymptote) provides information about the dynamics of transfer. In turn, how factors other than dose modulate the dynamics of transfer from CT can be examined by comparing D-R functions. Besides shedding light on transfer per se, a better understanding of D-R relations might aid in the interpretation of prior studies, and thus contribute towards a more integrated body of knowledge and greater consensus.

For example, differences in dosage might help to explain some of the disparate findings in the literature on the efficacy of CT. One noteworthy case involves the first large-scale study, performed by Owen et al. [16], of the efficacy of CT over the internet (results from 11,430 participants). This study found no difference in effects on a neuropsychological test battery between CT like that provided by commercial programs and an active control group. These findings, however, appear to be at odds with those from a subsequent large-scale study (6,742 participants recruited) by a team that included several members of the same research group. This latter study, by Corbett et al. [17], involved the same CT tasks, activity by the control group, and neuropsychological tests. Here, benefits of CT over the internet were found on the neuropsychological tests, as well as on an additional self-report survey on instrumental activities of daily living.

Corbett et al. [17] suggested two possible explanations for the different outcomes of the two studies. One concerned differences in age between participants in the two studies. Participants were between 18 and 65 in the Owen et al. [16] study but were all over 50 in the Corbett et al. study. Corbett et al. suggested that, as a result of cognitive aging, CT might be more efficacious for older individuals. Evidence against this suggestion, however, is provided by another large online study of commercial CT. Analyzing results from 4,715 participants aged 18 to 85,

Hardy et al. [18] found benefits of CT relative to an active control on both a neuropsychological test battery and survey of real-world cognition and affect. Moreover, further analysis of this dataset by Ng et al. [19] found no difference in size between the CT effect for participants older and younger than 50.

The other possible explanation suggested by Corbett et al. [17] involved differences in CT dosage. Owen et al. [16] provided six weeks of training, while Corbett et al. provided six months. Moreover, Corbett et al. note that during the first six weeks of their study, participants engaged on average in approximately twice as much training as those in the Owen et al. study. The average total amount of training in the Owen et al. study consisted of 24 3-minute sessions, with a minimum of two sessions. Total training time for the entire cohort is not provided by Corbett et al., but it was clearly more than in the Owen et al. study. Participants in the Hardy et al. [18] study trained on average a total of 15 hours over a 10-week period.

A positive relation between dose and the size of a treatment effect might seem natural. Moreover, in the case of CT, one might expect this relation to be monotonic. But, surprisingly, such a relation has been challenged by results reported in meta-analyses combining the outcomes of a large number of studies. Specifically, some analyses investigating what study-design features moderate the size of CT treatment effects have failed to show larger effects in those studies with more training. A noteworthy example is a meta-analysis by Lampit et al. [10] of 52 RCTs of CT efficacy involving 4,885 cognitively healthy older adults. While there were overall effects of CT, as measured by neuropsychological tests of cognition, the size of the effects was not found to differ between studies employing more or less than 20 hours of total training. A more recent meta-analysis by Lampit et al. [11] also failed to observe a dose-response relation. This later analysis involved 90 studies (including 51 from [10]) of 7,219 cognitively healthy older adults, employed more sensitive measures of moderator effects, and compared studies with three levels of training (less than 9, 9–20, and greater than 20 hours).

Indeed, at least one meta-analysis found evidence suggesting that more CT can result in less transfer to neuropsychological tests. In a meta-analysis of 20 studies involving 913 cognitively healthy adults on the efficacy of both CT and non-CT video games, Toril et al. [12] found larger effects in studies with one to six weeks of training than in studies with seven to 12 weeks. It is unclear, however, how well the number of weeks spent training in each of these studies corresponds to the total amount of training, e.g., total duration or total number of games. Moreover, two studies in the group with fewer weeks of training did not include control groups, which would be expected to inflate the size of their treatment effects (Eq 1 vs. 2 in [12]).

These and findings from other meta-analyses about the effects of CT dosage should be interpreted with caution. While analyses of moderating factors can provide valuable information, they do have a number of limitations [23]. Even in meta-analyses involving a large number of studies and participants, high levels of sensitivity are difficult to obtain. Moreover, differences between studies on factors different from, but correlated with, the one on which they are being compared can also be a problem. Besides comparing the effects of different amounts of CT between studies, it would therefore be useful to examine the dose-response relation for CT within a single study, preferably one involving a sufficient number of participants to provide a high level of sensitivity.

We report here the results of a large observational study that involved approximately 107,000 users of Lumosity, a commercial program comprising computer games designed to provide CT over the internet. In addition to training with Lumosity games, these users elected to take an online battery of cognitive assessments (the NeuroCognitive Performance Test, NCPT) on two or more occasions separated by at least 10 weeks. Changes in performance on the NCPT between the first and second assessments were examined as a function of the

amount of intervening gameplay. Because the amount of gameplay between assessments differed across users, it was possible to calculate dose-response functions relating amount of gameplay to change in NCPT performance. D-R functions were obtained both for overall performance on the NCPT and for performance on its individual subtests, which are more directly related to specific cognitive domains. We examined also the differences between D-R functions from different demographic groups defined by age, gender, and education. The results of this study provide precise information about the form of the D-R function for CT and how it depends on a number of germane factors, including the course of adult aging.

## Methods

### Ethics statement

Both the study and release of the de-identified data were determined by WCG IRB (www.wcgirb.com) to be exempt research under 45 CFR § 46.104(d)(4) because 1) participants' identities could not be readily ascertained, 2) the investigator did not contact participants, and 3) participants would not be re-identified. Use of de-identified data for any purpose is also covered in the privacy policy agreed to by all participants during their registration to use the Lumosity CT program (www.lumosity.com/legal/privacy_policy). Although the study was determined to be exempt research and participants had already agreed to the use of their deidentified data, participants were additionally informed online immediately prior to the administration of each assessment battery that their data would be used for research purposes. To proceed with the assessment, participants clicked an "I Agree" button.

### Design and procedure

The procedure is outlined in Fig 1. A subset of Lumosity subscribers were invited via email and an in-app prompt to take the NCPT battery (Time 1, T1). Taking the NCPT was optional and not required for continued use of Lumosity. After taking the NCPT battery at T1, participants were locked out of the battery until invited 70 days later via email and an in-app prompt to take it a second time (Time 2, T2). Between assessments, participants could continue to freely play the games included in their subscription (described below). Importantly, the games were distinct from the NCPT subtests. The dose-response functions examined in this study express the relation between the change in NCPT performance from T1 to T2 and the number of games played during the interim.

### Participants

All participants were recruited from Lumosity subscribers who signed up from 3-26-2007 to 3-26-2020 and indicated English as their preferred language. Country of origin was restricted to the US, Canada, Australia, and New Zealand. NCPT assessments were completed at both T1 and T2 by 213,569 participants. To exclude individuals with substantial training before their T1 (baseline) assessment, a threshold was set on the amount of gameplay between initial registration and this assessment. Based on a criterion of 25 games or less, results from 107,005 participants were fully analyzed (for details and rationale, see first section in Results).

The age, gender, and educational composition of the fully analyzed participants are described in Table 1 and Fig 2. All participants reported their age, but some did not report a gender (9.41%) or classifiable level of education (11.46%). The age of participants ranged from 18 to 89, with a mean of 51.7. More participants reported being female (55.35%) than male (35.24%). Overall, participants were highly educated, with 26.58% reporting graduate or professional degrees and only 9.07% reporting a high school degree or less. Overall, the women

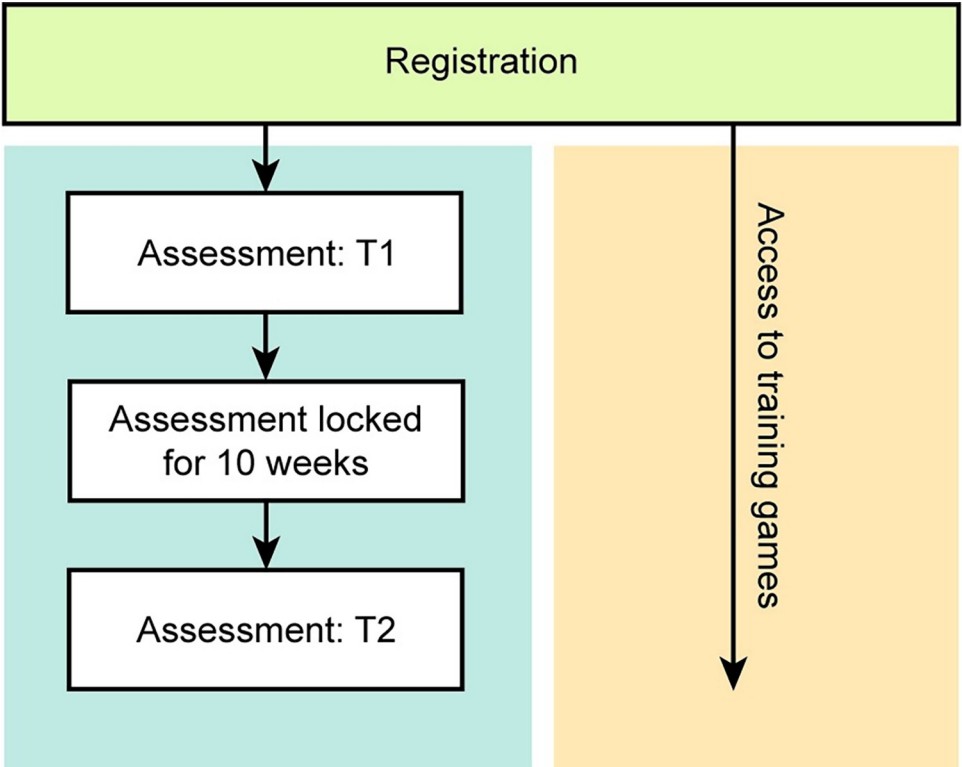

**Fig 1. Procedure.** After registering for the Lumosity program (green), participants were eligible for assessment with a neuropsychological test battery (NCPT) at two time points separated by at least 10 weeks (blue). In parallel, participants freely played games from the Lumosity CT program (yellow), with the amount of gameplay varying between participants. Changes in performance on the NCPT between assessments were examined as a function of the amount of intervening gameplay.

were older than the men (53.34 vs 48.68, t = 46.504, p (two-tailed) < 0.0001). As can be seen from Fig 2, this was due in part to the large representation of middle-aged and older women in the cohort. Though the median level of education for both genders was a Bachelor's degree, the women were more educated overall than the men (W = 1,013,742,340, n1 = 57,125, n2 = 36,437, p (two-tailed) < 0.0001). As might be expected, age and educational level were positively correlated (rho (94,740) = 0.073, p < 0.0001).

## Cognitive training

CT was provided via the Lumosity program, which included 69 different games (each described in S1 Appendix) across web and mobile apps over the time period in which the study data were collected. (Games were introduced and discontinued over time, so the exact number of available games varied.) Lumosity games are modeled on paradigms used to study specific cognitive functions in the lab or clinic. Based on their primary cognitive demands, they can be organized into seven cognitive domains: Memory, Attention, Flexibility, Problem Solving, Speed of Processing, Math Skills, or Language Skills. The majority of games, however, make demands in multiple cognitive domains (e.g., Speed of Processing and Flexibility, or Attention and Memory). As members of Lumosity, participants could train whenever and as much as they pleased. They could play any of the games available on their platform. However, to encourage breadth of training, each new day a user logged onto the platform, they were recommended five particular games. A single five-game session typically lasted about 15 minutes.

**Table 1. Age, gender, and education of fully analyzed participants.**

| Gender | | | | |
|---|---|---|---|---|
| **Education (count, % total)** | **Female** | **Male** | **Unreported** | **Total** |
| **HS grad or less** | 5,506 | 4,104 | 95 | 9,705 |
| | 5.15 | 3.84 | 0.09 | 9.07 |
| **Some college** | 14,605 | 9,337 | 303 | 24,245 |
| | 13.65 | 8.73 | 0.28 | 22.66 |
| **Bachelor's degree** | 19,530 | 12,415 | 405 | 32,350 |
| | 18.25 | 11.60 | 0.38 | 30.23 |
| **Advanced degree\*** | 17,484 | 10,581 | 377 | 28,442 |
| | 16.34 | 9.89 | 0.35 | 26.58 |
| **Other/unreported** | 2,100 | 1,271 | 8,892 | 12,263 |
| | 1.97 | 1.19 | 8.31 | 11.46 |
| **Total** | 59,225 | 37,708 | 10,072 | 107,005 |
| | 55.35 | 35.24 | 9.41 | 100.00 |
| **Age (years)** | | | | |
| **Mean** | 53.34 | 48.68 | 53.35 | 51.7 |
| **SD** | 14.29 | 16.57 | 15.08 | 15.37 |
| **Range** | 18–89 | 18–89 | 18–89 | 18–89 |

\*Masters, Ph.D., or professional degree.

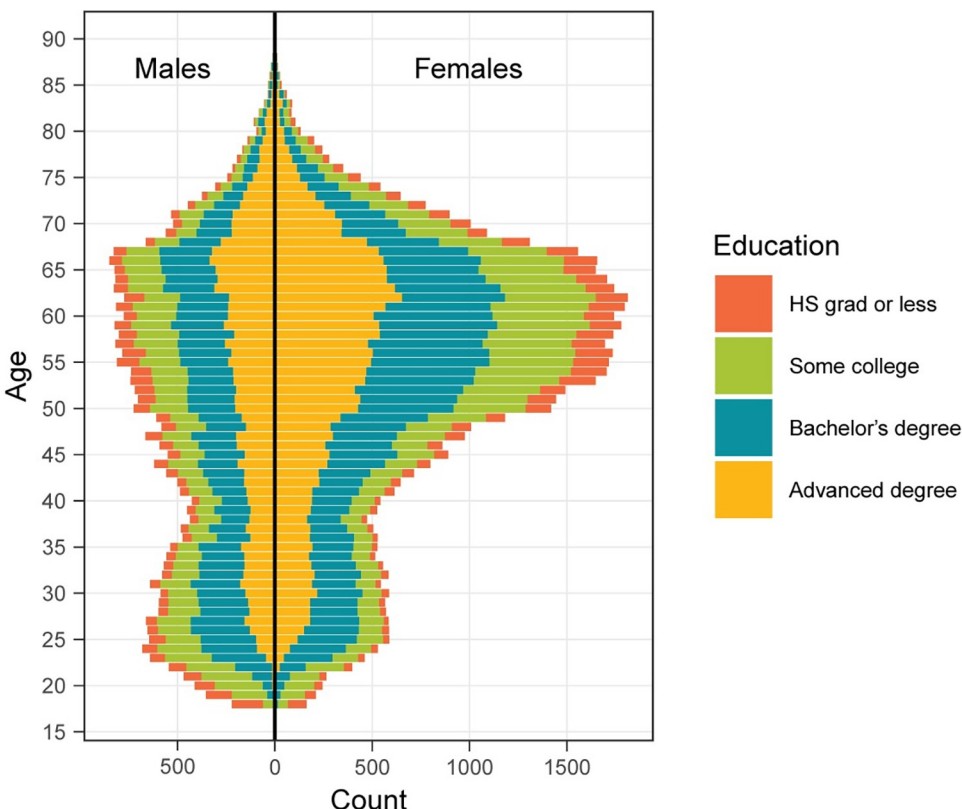

**Fig 2. Population pyramid for fully analyzed participants reporting age, gender, and a classifiable level of education (87%, N = 93,562).**

## Neuropsychological assessment

The NeuroCognitive Performance Test [24, 25] is a validated, brief, repeatable, web-based cognitive assessment platform that measures performance across multiple cognitive domains. NCPT subtests are digital translations of commonly used neuropsychological assessments, which can be selected and arranged to form customized batteries. The specific battery used here took about 20–30 minutes to complete and included the eight subtests described below in their order of presentation. (More detailed descriptions are provided in S2 Appendix).

(1) Arithmetic Reasoning: Designed to assess numerical problem-solving ability [26]. Participants are required to respond as quickly and accurately as possible to arithmetic problems written in words (e.g., "Four plus two ="').

(2) Digit Symbol Coding: Used to measure speed of processing and memory [27]. Participants enter the number corresponding to the symbol using the key provided at the top of the screen.

(3) Forward Visual Memory Span and (4) Reverse Visual Memory Span: Based on the Corsi Blocks tasks [28] and designed to assess visual short-term and working memory respectively. Participants are required to recall a sequence of randomized spatial locations in either forward or reverse order.

(5) Trail Making A: Used to measure speed of processing [29]. Participants connect the numbers from smallest to largest as quickly as possible.

(6) Trail Making B: Used to measure speed of processing and mental flexibility [29]. Subjects connect numbers (from smallest to largest) and letters (in alphabetical order) alternating between the two (i.e., 1 to A to 2 to B to 3 to C, etc.).

(7) Grammatical Reasoning: Based on Baddeley's Grammatical Reasoning Test [30] and designed to assess cognitive flexibility and reasoning. This subtest requires participants to rapidly and accurately evaluate potentially confusing grammatical statements.

(8) Progressive Matrices: Based on established matrix reasoning assessments [31] and designed to assess problem solving and fluid reasoning.

## Outcome measures

Each study participant's score on each subtest at T1 and T2 was scaled using a pre-computed norm table that mapped raw scores to values on a normal distribution with a mean of 100 and SD of 15. The norm tables were calculated separately for each subtest using scores on their first NCPT from a larger set of Lumosity users who met the following criteria: completed the entire test battery, had logged no more than 25 prior gameplays, and reported their age and educational attainment. The ranks of the scores in each table were reweighted to account for demographic differences (age and educational attainment) between the normative dataset and the 2019 US Census Bureau's American Community Survey (ACS) 1-year Public Use Microdata Sample. Thus, the ranks in the norm tables approximated those from a population with the same demographic composition as that reported in the 2019 ACS Microdata Sample. Finally, based on the percentile of its rank, each score in the norm table for each subtest was converted by inverse-normal transformation to a value in a normal distribution with a mean of 100 and SD of 15. After scaling with the norm tables, the mean of the scaled subtest scores for each participant was used to generate an overall composite score (Grand Index, GI) on the T1 and T2 NCPT assessments using an analogous norm table. For further details, see [24, 25].

## Statistical software and fit models

All statistical analyses were performed using R statistical software [32], version 4.0.0. Exponential curves were fit to D-R functions using the R nls function. Analyses based on bootstrapping employed the boot and boot.ci functions in the R boot package. Linear regressions employed the R lm function. ANOVAs employed lm and the Anova function in the R car package. Type I errors resulting from multiple within-family tests were controlled for using the Benjamini-Hochberg Procedure [33], which was performed by the R p.adjust function.

Many of the analyses involved examining the parameters of exponential functions fit to D-R functions. In each, the parameters of one or more exponential functions were estimated by fitting a single model to data involving number of gameplays and changes on the NCPT. The three models employed here are all based on Eq 1 (Results), which defines an exponential approach to an asymptote. Among the variables in each are one or more intercepts (I), asymptotes (A), and rates (R). The models are described below in order of their complexity.

**Model 1.** The simplest model fit a single exponential function to data from the entire set of approximately 107K fully analyzed participants.

$$\Delta Y_i = I + (A - I)*(1 - e^{-R*N_i})$$

Here the subscript i refers to a given participant. $\Delta Y$ refers to their change on the NCPT, either in overall score (GI) or on one of the eight subtests. N is the number of games they played between the two NCPT assessments. The model is fit for GI in the Results section "Characterization of D-R functions" (see Fig 5) and for individual subtests in the Results section "D-R functions for individual subtests of the NCPT" (see Figs 8, 9 and Table 5).

**Model 2.** A more complex model simultaneously estimates the intercept, asymptote, and rate for each of four levels of the demographic factor AGE (3 x 4 = 12 parameters). The model was fit to data from the approximately 93.5K fully analyzed participants with complete demographic information. Besides exponential parameters, the model includes dummy variables that depend on the age group to which each participant belongs. The rationale for Model 2, as opposed to applying Model 1 separately to each age group, was to estimate the exponential parameters more precisely by pooling data across all four age groups in the same model.

$$\Delta Y_i = I^* + (A^* - I^*)*(1 - e^{-R^**N_i})$$

where

$$I^* = I_1 + \sum_{j=2}^{4} IS\_AGE_{ij}*I_j$$

$$A^* = A_1 + \sum_{j=2}^{4} IS\_AGE_{ij}*A_j$$

$$R^* = R_1 + \sum_{j=2}^{4} IS\_AGE_{ij}*R_j$$

Again, i indexes participants, $\Delta Y$ is change on the GI or one of the subtests, and N is number of games between the two NCPT assessments. Age group is indexed by j (1 to 4) and determines the values (0 or 1) of three dummy variables ($IS\_AGE_{ij}$, j = 2 to 4) for each participant. Here one age group (youngest, j = 1) is treated as a reference group, with estimates of its intercept, asymptote, and rate parameters ($I_1$, $A_1$, and $R_1$) expressed in absolute terms. Exponential

parameters for the other three age groups ($I_{2 \text{ to } 4}$, $A_{2 \text{ to } 4}$, and $R_{2 \text{ to } 4}$) are expressed relative to the reference group values. This model is fit for GI in the Results section "Age differences in D-R functions" (see Table 4 and Figs 6 and 7) and for individual subtests in the Results section "Age differences on individual subtests of the NCPT" (see Fig 10).

**Model 3.** The most complex model incorporated three demographic factors: 4 levels of EDU, 2 levels of GEN, and 4 levels of AGE. Like Model 2, it was fit to data from the fully analyzed participants with complete demographic information and pools data across demographic groups. This model includes an estimate in absolute terms of an intercept, asymptote, and rate parameter for a single reference group (youngest AGE, high school or less EDU, female GEN). Exponential parameters estimated for the other levels of each demographic factor are expressed relative to the reference group. This resulted in 24 parameters [3 (I, A, R) x 8 (1 ref + 3 non-ref AGE + 3 non-ref EDU+ 1 non-ref GEN)]. Also included were dummy variables that depended upon the demographic classification of each participant.

$$\Delta Y_i = I^* + (A^* - I^*) * (1 - e^{-R^* * N_i})$$

where

$$I^* = I_{111} + \sum_{j=2}^{4} IS\_AGE_{ij} * I_j + \sum_{k=2}^{4} IS\_EDU_{ik} * I_k + \sum_{m=2}^{2} IS\_GEN_{im} * I_m$$

$$A^* = A_{111} + \sum_{j=2}^{4} IS\_AGE_{ij} * A_j + \sum_{k=2}^{4} IS\_EDU_{ik} * A_k + \sum_{m=2}^{2} IS\_GEN_{im} * A_m$$

$$R^* = R_{111} + \sum_{j=2}^{4} IS\_AGE_{ij} * R_j + \sum_{k=2}^{4} IS\_EDU_{ik} * R_k + \sum_{m=2}^{2} IS\_GEN_{im} * R_m$$

Again, $\Delta Y$ refers to change on the NCPT and N indicates the number of games between NCPT assessments. The subscripts i, j, k, and m index participant, AGE group, EDU group, and GEN. $IS\_AGE_{ij}$, $IS\_EDU_{ik}$, $IS\_GEN_{im}$ are dummy variables that depend on a participant's group membership. $I_{111}$, $A_{111}$, and $R_{111}$ are the intercept, asymptote, and rate parameters for the reference group (j, k, and m = 1). Exponential parameters are indexed by j = 2 to 4 for the other levels of AGE, by k = 2 to 4 for the other levels of EDU, and m = 2 for the other level of GEN. This model is fit for GI in the Results section "Comparison of D-R functions across age, education, and gender" (see Table 3).

The parameter estimates resulting from each fit of each of the three models were analyzed statistically using bootstrap procedures. Bootstrap sampling enabled the computation of confidence intervals and the performance of statistical tests by providing a source of replication for the parameter estimates without requiring assumptions about their distributions. Individual statistical analyses and associated bootstrap procedures are described in the Results section.

## Results

### Preliminary analyses: Influence of prior CT on subsequent D-R functions

The D-R functions examined here express the relation between the change in NCPT performance from T1 to T2 and the number of games played during the interim. This relation, however, is influenced by the amount of training on the games before T1: More gameplay before T1 was associated with smaller effects of subsequent gameplay on NCPT performance. As preliminary steps, we therefore sought to characterize, explain, and then minimize this influence.

Fig 3 shows the change in GI from T1 to T2 as a function of both the amount of gameplay between T1 and T2 (rows) and the amount of gameplay before T1 (columns) for the 213,568 participants who took the NCPT twice. With low levels of pre-T1 gameplay (leftmost columns), there was a strong positive relation between the amount of T1-to-T2 gameplay and change in GI; With high levels of pre-T1 gameplay (rightmost columns), increases in T1-to-T2 gameplay led to little or no increase in GI. Consistent with this pattern, a multiple linear regression of change in GI on the logarithms (log1p) of pre-T1 and T1-to-T2 gameplay included a negative interaction term (t(213,565) = -6.707, p < 0.0001). The regression term was positive (t(213,565) = 14.620, p < 0.0001) for T1-to-T2 gameplay and non-significant (t(213,565) = 0.364, p = 0.7161) for pre-T1 gameplay.

The influence of gameplay (CT) prior to T1 on the D-R function can be explained by the combination of two effects. First, CT improves performance on the NCPT at T1. The linear regression of GI at T1 on the logarithm (log1p) of the number of gameplays before T1 has a positive slope (t(213,567) = 46.66; p < 0.0001). Second, there are diminishing returns in the effects of CT on subsequent NCPT performance (hence the log terms in the above linear regressions). These are implied by the form of the D-R function, which is examined in the next section. Because of such diminishing returns, the more CT prior to T1 improves NCPT performance at T1, the less any given amount of CT between T1 and T2 will further improve NCPT performance at T2.

To minimize the effects of prior CT on the D-R functions, further analyses included only those participants (N = 107,005) with no more than the median number (25) of gameplays before T1. Since each of the columns in Fig 3 corresponds to a tenth of the participants, the

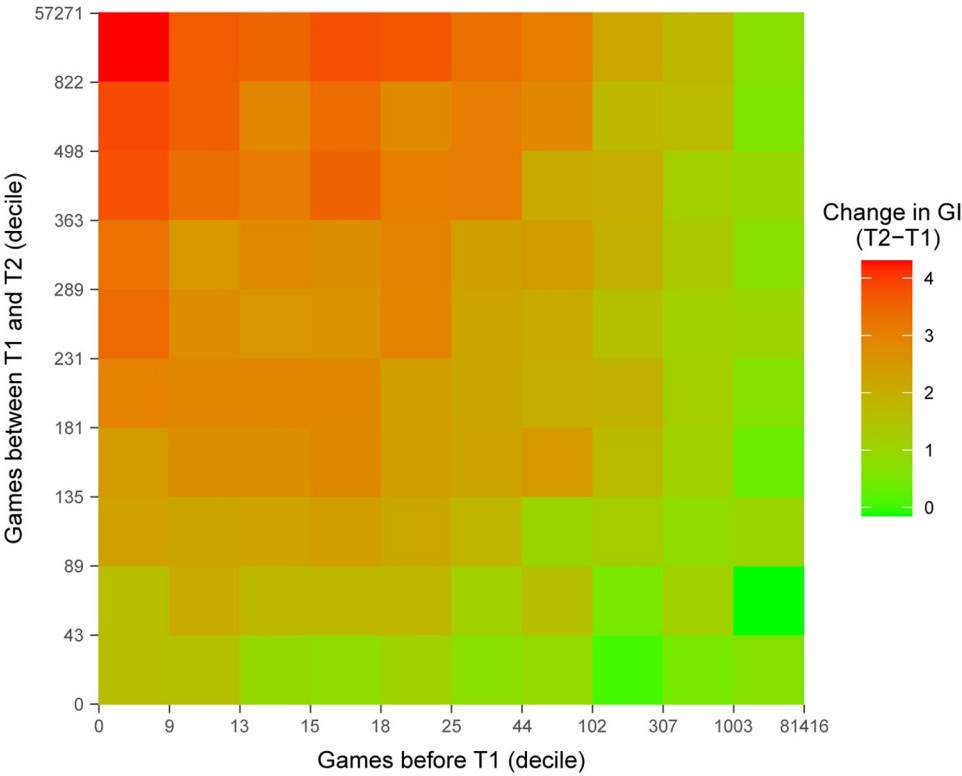

**Fig 3. Change in GI from T1 to T2 as a function of the amount of gameplay between T1 and T2 and amount of gameplay before T1.** Amounts of pre-T1 and T1-to-T2 gameplay are shown respectively on the x- and y-axes. The numbers on each axis are the upper limits of 10 bins of participants partitioned by number of games played. The amount of change in GI from T1 to T2 for each combination pre-T1 and T1-to-T2 gameplay is indicated by color.

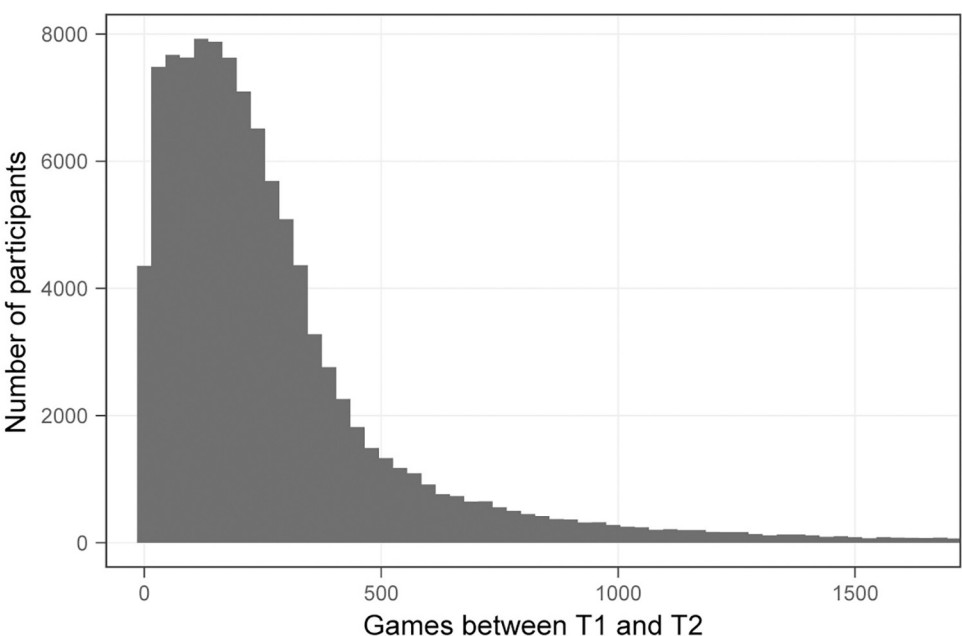

**Fig 4. Distribution of T1-to-T2 gameplays for fully analyzed participants.** Note that the distribution is truncated at the 98[th] percentile.

included participants are those who contributed to the five leftmost columns. As mentioned in the Methods, this is the set of participants whose demographics are described in Table 1 and Fig 2.

Fig 4 shows the distribution of the number of T1-to-T2 gameplays up to the 98[th] percentile (1642) for these fully analyzed participants. The distribution ranged from 0 to 21,689 gameplays, with a mean of 321 and median of 208. The resulting D-R function for overall performance on the NCPT is examined next.

## Characterization of the D-R function: Empirical form and exponential fit

The D-R function involving all fully analyzed participants (25 or less gameplays before T1) is presented in Fig 5. Again, dose is expressed in terms of the number of games played between T1 and T2 and the response is expressed in terms of GI (overall performance on the NCPT). Note that the dose metric can be converted to number of hours of CT by dividing by 20, since the average game duration was about 3 minutes. (A similar plot in which the response metric is effect size, expressed in units of Hedges' g, can be found in S1 File). The dots show means for 20 equal-sized bins of participants partitioned by dose, and the line shows an exponential function fit to the 107,005 points corresponding to individual participants. The heights of the blue and yellow areas represent effects on the D-R function from the two different sources of learning discussed below.

Among the salient features of the figure are three that can be interpreted in terms of learning on the NCPT and transfer to the NCPT from CT. One is the y-intercept (0 gameplays between T1 and T2), which indicates improvement on the NCPT due to repeated testing. Based on participants with minimal (25 or less) gameplays between T1 and T2, the mean of this effect is 1.529 ($t(6,764) = 17.015$, $p < 0.0001$) with a 95% CI of 1.353 to 1.705. Another feature is the presence of transfer over and above the effect of repeated testing: Positive change in NCPT performance increases with increasing CT between T1 and T2. Based on a comparison

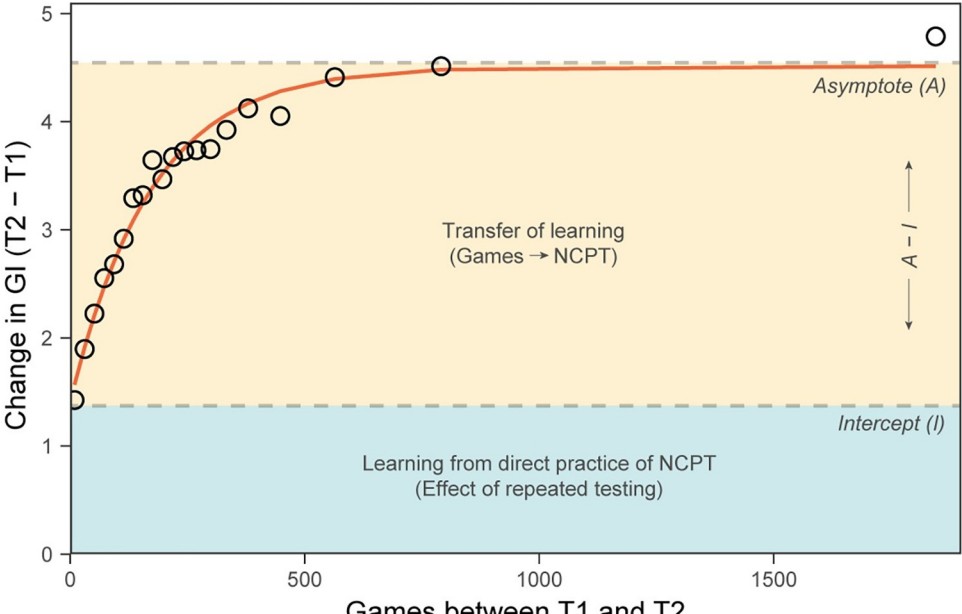

**Fig 5. Empirical D-R function (dots) and exponential fit (line) for NCPT Grand Index (GI) involving all fully analyzed participants.** Dots show means for 20 equal-sized bins of participants partitioned by number of games played between T1 and T2. The line shows an exponential approach to an asymptote (Eq 1) fit to the 107,005 points corresponding to individual participants. The heights of the blue and yellow areas indicate respectively the effect of repeated testing and the maximum effect of transfer from gameplay on NCPT performance.

between participants with 25 or fewer gameplays and those with 1000 or more gameplays, the mean of this effect is 3.273 ($t(11,398) = 24.477$, $p < 0.0001$) with a 95% CI of 3.011 to 3.536. The size of the effect (Hedges' g) is 0.449 with a 95% CI of 0.412 to 0.485. A third feature concerns the form of the D-R function. With increasing CT, change in NCPT performance is most rapid initially, decelerates, and then levels off approximating an asymptote. This indicates that, as mentioned in the previous Results section, CT has diminishing returns in its effects on subsequent NCPT performance.

Moreover, as shown in Fig 5, the form of the empirical D-R function can be well approximated by an exponential approach to an asymptote (Eq 1).

$$\text{change on NCPT} = a + (b - a) * (1 - e^{-c*\text{games between T1 and T2}}) \tag{1}$$

The fit exponential is useful because its three parameters (a, b, and c) also estimate learning on and transfer to the NCPT. The y-intercept (a = 1.403) estimates the effect on the NCPT of repeated testing and is close to the mean value calculated directly from the data (empirical D-R function). The difference between the asymptote and y-intercept (b–a = 4.513–1.403 = 3.11) estimates the maximum effect of transfer from CT on NCPT performance and is also close to the mean value calculated from the empirical D-R function. The rate parameter (c = 0.0058) is inversely proportional to the number of gameplays necessary to produce any given level of change on the NCPT. For example, it indicates that about 63% (1–1/e) of the growth in the fit function (Fig 5) occurred by the 172[nd] (1/0.0058) gameplay.

In this paper, we examine how D-R functions are influenced by several factors. Specifically, we sought to determine whether these factors influence the effect of repeated testing or amount of transfer by examining their influence on parameters of exponential fits to D-R functions

(see Statistical software and fit models in Methods). Effects of demographic factors will be examined first.

## Comparison of D-R functions across age, education, and gender

Here we determine whether D-R functions vary with age, education, or gender and, if so, which features of the fit exponential function (intercept, rate, or asymptote–intercept) are involved. The participants examined are those with 25 or fewer gameplays before T1 who also provided complete demographic information (N = 93,562). The factor levels employed for education (HS degree, some college, college degree, and advanced degree) and gender (female and male) are those described in the Methods. The continuous age factor described in the Methods is divided here into four levels: 18–40 (N = 23,276), 41–55 (N = 26,727), 56–70 (N = 35,510), and 71–89 (N = 8,049).

First, to see whether D-R functions differed in any respect across each demographic factor, we performed a multiple linear regression that compared the overall amount of change in GI between demographic levels, while controlling for differences in amount of gameplay. The need for such control was examined in an additional analysis, which found that the amount of gameplay did in fact differ across the levels of each demographic factor. An ANOVA (with Type 2 SSs to control for correlations between factors) found significant effects on number of games played of Age ($F_{(3,93554)}$ = 74.012, $p < 0.0001$), Education ($F_{(3,93554)}$ = 32.611, $p < 0.0001$), and Gender ($F_{(1,93554)}$ = 215.512, $p < 0.0001$). The linear regression of demographic effects on overall change in GI therefore included the number (on a log scale) of each participant's gameplays between T1 and T2 as a covariate. The results are shown in Table 2. As can be seen, overall change in GI between T1 and T2 differed between Age levels ($F_{(3,93553)}$ = 79.592, $p < 0.0001$), but not between levels of Education ($F_{(3,93553)}$ = 1.221, $p = 0.3003$) or Gender ($F_{(1,93553)}$ = 0.626, $p = 0.4287$). Note also the significant ($p < 0.001$) increase in GI change scores with number of games.

Next, to specify further how D-R functions vary with age, as well as confirm the absence of differences across education and gender, we examined how the parameters of fit exponential functions varied across the three demographic factors. This involved fitting a single model that included an intercept, asymptote–intercept, and rate parameter for each combination of

**Table 2. Linear regression of change in NCPT Grand Index on demographic factors while controlling for number of games played.**

| Model: GI_Change ~ log1p(Num_Games) + Age + Education + Gender | | | |
|---|---|---|---|
| **Coefficients** | **Estimate** | **Std. Error** | **t value** | **Pr(>|t|)** |
| Intercept (0 games,18–40, HS, F) | 0.22966 | 0.12965 | 1.771 | 0.0765^ |
| log1p(Num_Games) | 0.70201 | 0.01896 | 37.032 | <2e-16*** |
| Age2 (41–55) | -0.51543 | 0.06230 | -8.274 | <2e-16*** |
| Age3 (56–70) | -0.74400 | 0.05904 | -12.601 | <2e-16*** |
| Age4 (71–89) | -1.18688 | 0.08955 | -13.254 | <2e-16*** |
| Education 2 (some college) | -0.02859 | 0.08319 | -0.344 | 0.7311 |
| Education 3 (college degree) | 0.07818 | 0.08018 | 0.975 | 0.3295 |
| Education 4 (advanced degree) | 0.04942 | 0.08149 | 0.607 | 0.5442 |
| Gender (male) | 0.03709 | 0.04686 | 0.791 | 0.4287 |

Significance

***p < 0.001

**p < 0.01

*p < 0.05, ^p < 0.1

**Table 3. Intercept, rate, and asymptote–intercept of fit exponential functions for each level of each demographic factor.**

| | Intercept (GI points) | Rate (game$^{-1}$) | Asymptote—Intercept (GI points) |
|---|---|---|---|
| Reference (18–40, HS, F) | 2.2286** | 0.00379** | 3.1486** |
| | *The following values are relative to the Reference group* | | |
| Age 2 (41–55) | -0.5612* | -0.00015 | 0.1192 |
| Age 3 (56–70) | -0.7209** | -0.00033 | 0.0511 |
| Age 4 (71–89) | -1.6260** | 0.00203 | 0.2857 |
| Education 2 (some college) | -0.3876 | 0.00194 | -0.0048 |
| Education 3 (college degree) | -0.3562 | 0.00311 | -0.1456 |
| Education 4 (advanced degree) | -0.4419 | 0.00311 | 0.0109 |
| Gender (male) | 0.1629 | -0.00067 | -0.0620 |

Significance

**$p < 0.01$

*$p < 0.05$

demographic levels (Model 3, Methods). The results are shown in Table 3. Analogous to Table 2, the top row shows the parameter values for a reference group (women 18–40 with a HS education), and the remaining rows show effects involving each demographic factor. Significance levels were obtained by means of 1,000 bootstrap replications that each fit Model 3 to resampled data. Resampling was done with replacement and preserved the number of participants at each combination of demographic levels.

Each exponential parameter for the reference group was found to be significantly greater than zero. Consistent with the linear regression, none of the parameters differed significantly between the reference group and other levels of education or gender. Also consistent were the results for age, which show that the effects on GI change score in the linear regression were due solely to changes in the intercept of the D-R function. That the asymptote–intercept and rate parameters were greater than zero for the reference group, but did not differ significantly across any of the demographic factors, is of particular interest. It implies that the dose-response relation was positive at each combination of age, education, and gender.

The above two analyses provide converging evidence that the positive relation between dose and response cannot be explained by demographic differences between individuals with different amounts of training, as this positive relation occurs at each combination of demographic levels. Both analyses found change in GI (response) to be positively related to amount of CT (dose). Neither found significant differences between the D-R functions across different levels of education or gender. The differences in GI change found across age in the linear regression were shown in the exponential analysis to result solely from differences in the intercept of D-R functions, rather than from differences in asymptote–intercept or rate, thus preserving the positive relation between dose and response. This pattern of age differences is confirmed and explored further in the next section.

## Age differences in D-R functions

Given the absence of differences between D-R functions across education and gender, we decided to focus on age. Understanding how the dose-response relation for CT varies with age is important for a variety of reasons, including that the meta-analyses discussed earlier [10–12] examined this relation in older adults only. We therefore combined levels of education and gender and examined how the parameters of fit exponentials varied across the sole factor of age. This involved fitting a single model with 12 values: 3 exponential parameters x 4 age levels

**Table 4. Intercept, rate, and asymptote–intercept of fit exponential functions for each age level.**

| | Intercept (GI points) | Rate (game⁻¹) | Asymptote—Intercept (GI points) |
|---|---|---|---|
| Reference (18–40) | 2.0485** | 0.00522** | 3.0370** |
| | *The following values are relative to the Reference group* | | |
| Age 2 (41–55) | -0.6801** | 0.00065 | 0.1187 |
| Age 3 (56–70) | -0.8909** | 0.00074 | 0.0728 |
| Age 4 (71–89) | -1.6196** | 0.00157 | 0.3256 |

Significance

**$p < 0.01$

*$p < 0.05$

(Model 2, Methods). Confidence intervals for these values were obtained by means of 1,000 bootstrap replications that each fit Model 2 to resampled data. Resampling was done with replacement and preserved the number of participants at each age level.

Parameters of the exponential fit at each age level are shown in Table 4 and Fig 6. The results in both are consistent with the findings for age in Table 3. In Table 4, the reference level (18–40) differs significantly from zero for all three parameters, but significantly from the other age levels only for the intercept. Medians and 95% confidence intervals of the bootstrapped values at each age level are shown in the figure. Here the parameter values for non-reference levels are shown directly, rather than as differences from those at the reference level. As in Tables 3 and 4, the intercept can be seen to decrease with increasing age, while rate and asymptote–intercept remain relatively constant.

Fig 7 shows the exponential functions for each age level based on the parameter values presented in Table 4. As can be seen from the asymptotes, overall improvement on the NCPT diminished with age. This diminution is captured by the age differences in GI change found in the linear regression (Table 2). However, the differences in asymptote are nearly identical to the differences in intercept, as can be seen from the nearly identical asymptote–intercept parameters (Fig 6). Likewise, there were no statistically significant differences between rate parameters (Table 4). Hence, the four functions are close to vertical translations of one another. This vertical shift, which is the only difference apparent between D-R functions for the age groups, is due solely to the age differences in intercept.

As discussed previously, the intercept provides an estimate of the change in GI due to repeated testing on the NCPT, while the rate and asymptote–intercept estimate characteristics of the transfer from gameplay to the NCPT. That age affected the intercept, but not rate or asymptote–intercept, therefore implies that it modulated the effects of repeated testing but not of transfer. Converging evidence for this conclusion is provided by the pattern of age effects on features of the empirical (non-fit) D-R function that approximate its intercept and the total amount of transfer. Age affected the change in GI for all participants with 25 or fewer gameplays ($F(3,5823) = 9.108$, $p < 0.0001$), but not the difference in GI change between these participants and those with 1000 or more gameplays ($F(3,10493) = 0.220$, $p = 0.8825$). The conclusion is supported also by supplementary analyses in which age was treated as a continuous variable rather than divided into four separate categories (see S2 File).

A decrease with age in benefits from repeated testing would suggest the existence of age differences in a type of learning ability. But there is also a trivial alternative explanation for the observed decrease that involves variability in when participants took their second NCPT relative to their first. Specifically, if the intervals between the two NCPTs were longer for older participants, it would provide them with greater opportunity to forget anything learned from the

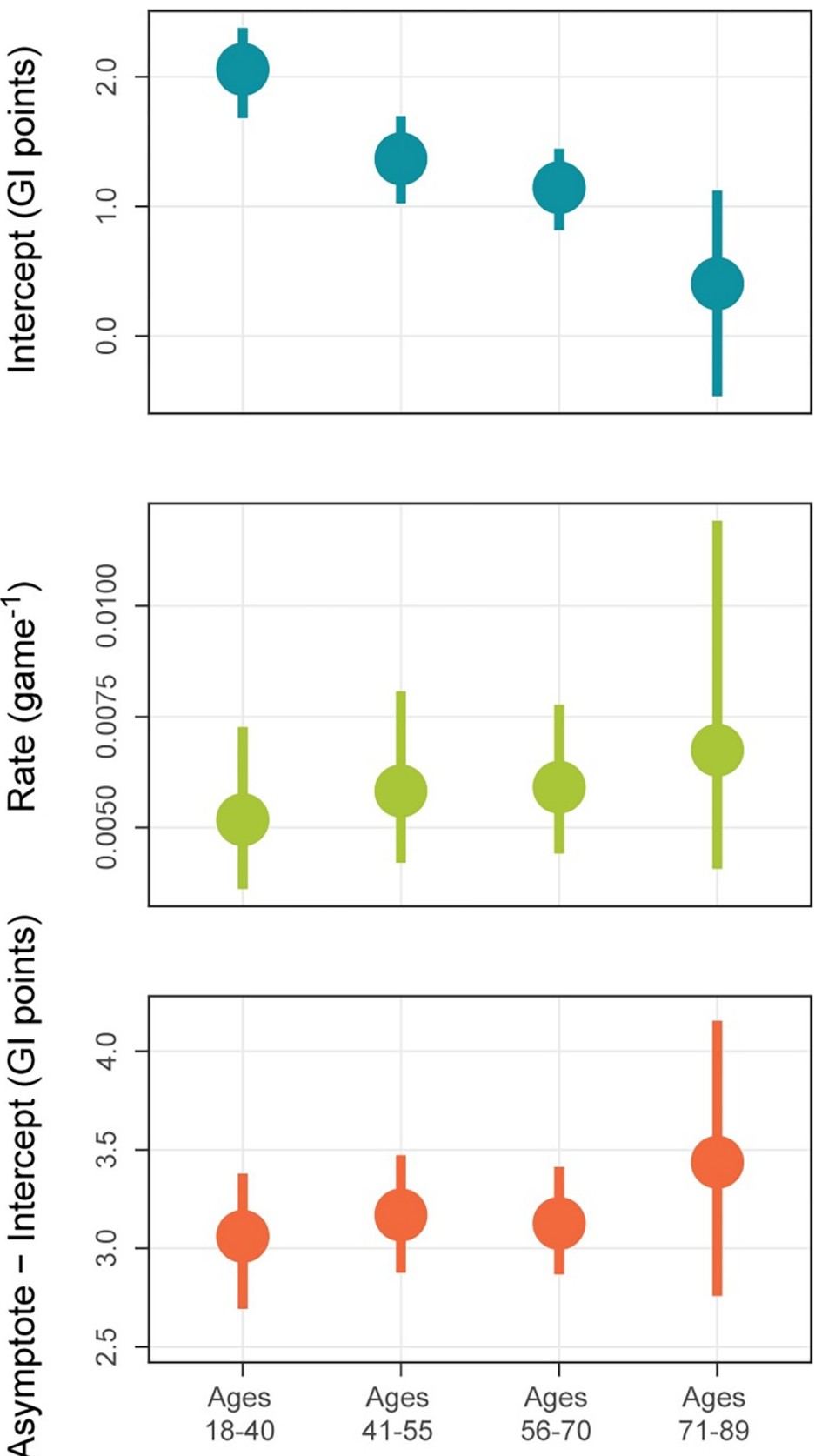

**Fig 6. Medians and 95% confidence intervals of intercept, rate, and asymptote–intercept of exponential functions fit to each age level.** Medians (center points) and CIs (upper and lower edges) were determined through a bootstrapping procedure.

first by the time of the second. This hypothesis can be rejected on the basis of the small negative correlation found between age and the T1-to-T2 interval (r(93,560) = -0.0846, p < 0.0001, 95% CI = -0.0910 to -0.0783). The direction of this correlation indicates a tendency towards shorter T1-to-T2 intervals, i.e., less opportunity to forget their first NCPT, for older than younger participants.

Also relevant to learning and cognitive aging is the absence of age differences in amount of transfer, especially when combined with the presence of age differences in the effects of repeated testing. The selective influence of age on the intercept, observed both on the empirical D-R function and fit exponential, suggests that the learning mechanism by which repeated practice on a task improves performance on the same task may not be identical to the mechanism by which practicing one task transfers to another. What the differences might be, and why the latter might be better preserved in adult aging, will be explored in the Discussion.

## D-R functions for individual subtests of the NCPT

So far, the examined D-R functions have described effects of transfer and repeated testing on GI, a measure of overall performance on the NCPT. As described in the Methods, GI is the scaled mean of scaled scores on eight individual subtests. Each subtest is a digital translation of a commonly used neuropsychological assessment. D-R functions for the individual subtests, which are more directly related to specific cognitive functions than GI, will be examined next. Because the scaling procedure provides a common metric for all the subtests, their respective D-R functions can be directly compared.

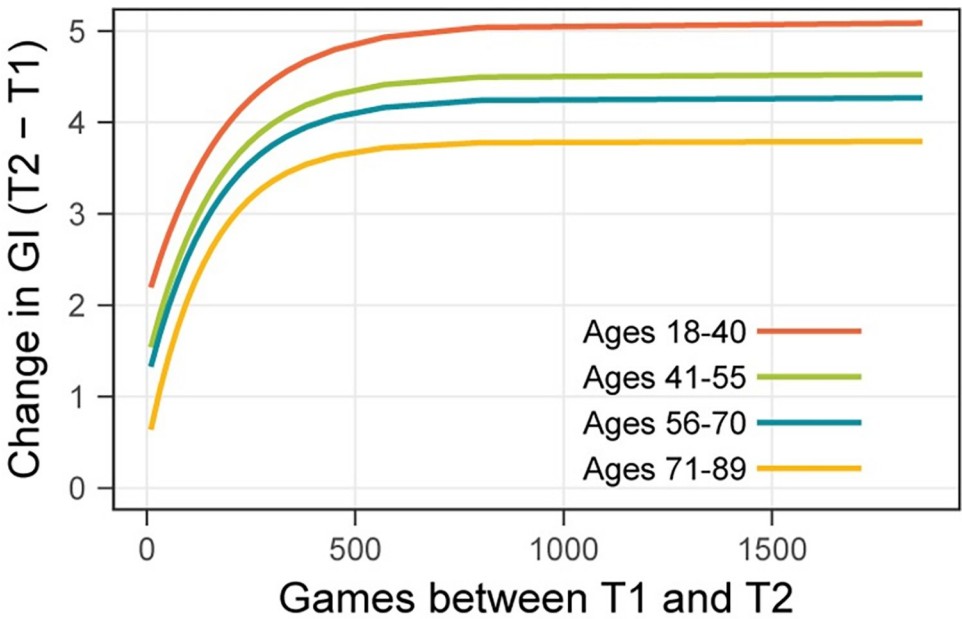

**Fig 7. Exponential approach to an asymptote fit to D-R function at each age level.** The intercept, rate, and asymptote—intercept are shown in Table 4.

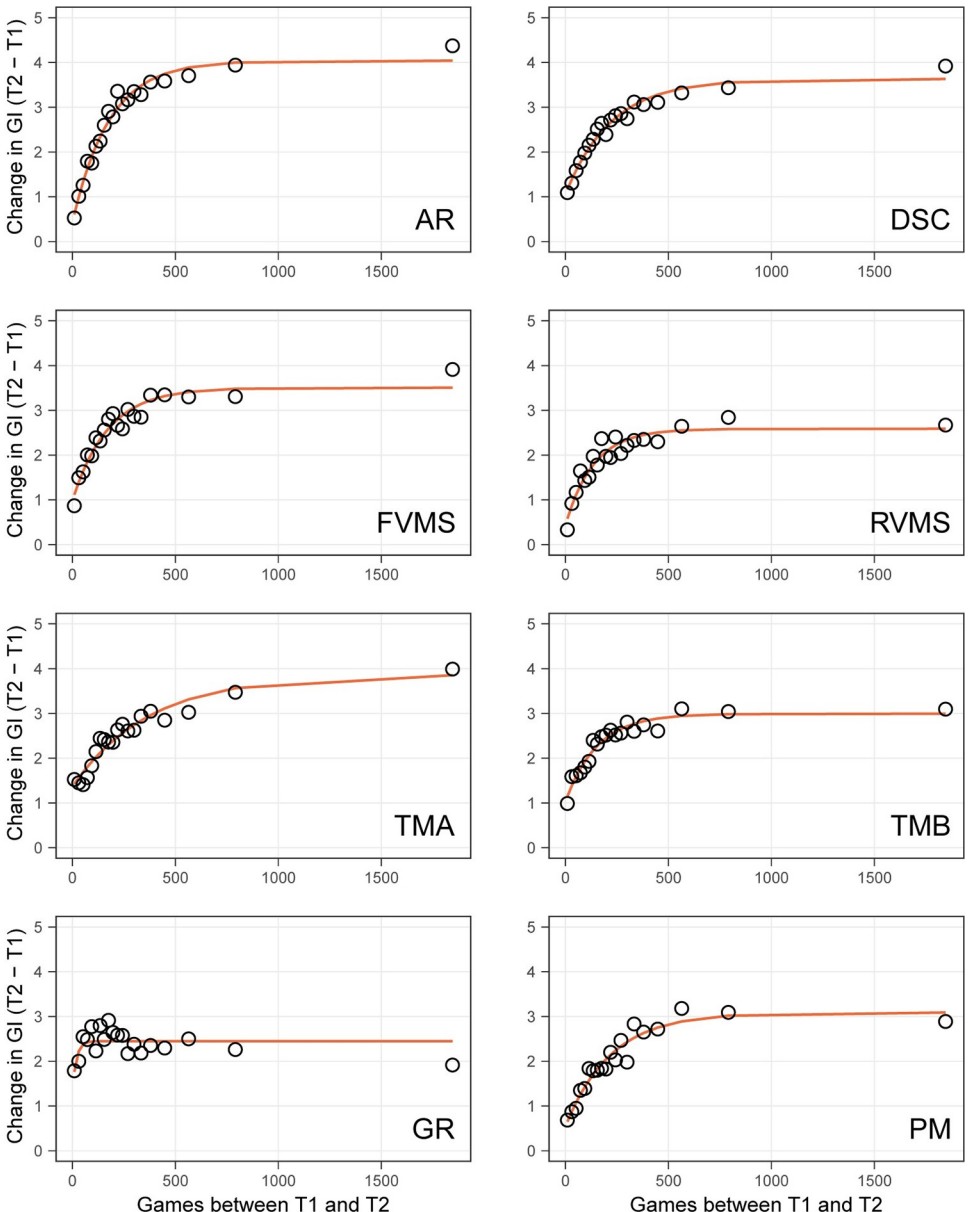

**Fig 8. Empirical D-R function (dots) and exponential fit (line) for each individual NCPT subtest.** Dots show means for 20 equal-sized bins of participants partitioned by the number of games played between T1 and T2. Lines show exponential functions (Eq 1) fit to the 107,005 points corresponding to individual participants. AR = Arithmetic Reasoning; DSC = Digit-Symbol Coding; FVMS = Forward Visual Memory Span; RVMS = Reverse Visual Memory Span; TMA = Trail Making A; TMB = Trail Making B; GR = Grammatical Reasoning; PM = Progressive Matrices.

Fig 8 displays the exponential fits to the D-R functions for each subtest. Here a model involving a single exponential function (Model 1, Methods) was applied separately to each individual subtest. The dots show means for 20 equal-sized bins of participants partitioned by number of games played, and the lines show exponentials fit to the 107,005 points corresponding to individual participants. As can be seen, the exponential function fits the D-R functions well for all subtests but Grammatical Reasoning (GR). Because of the poor fit, the parameter estimates for GR are questionable and not considered further. It should be noted, however,

that the empirical D-R function (dots) shows a large intercept and little growth with increasing CT, i.e., a large effect of repeated testing and little or no transfer.

Parameters of exponential fits for the remaining seven subtests are shown in Table 5. Also shown are features or components of cognitive function to which the subtests are thought to be especially sensitive (see Methods; S2 Appendix; [24, 25]). Of particular interest is whether the parameters differ between subtests and whether the pattern of such differences can be explained by the cognitive functions assessed.

Fig 9 shows 95% CIs for the parameter estimates displayed in the above table, as well as the statistical significance of contrasts between the different subtests. These were obtained by means of 1,000 bootstrap replications that each fit exponentials to resampled data. Resampling was done with replacement, and in such a way that the exponentials fit to the seven subtests on a given iteration were all based on the same sample of participants. This latter feature is important because Subtest was a within-subject factor. (While the repeated measurements made of the same participant has no impact on the CIs for the individual subtests, it does affect contrasts between subtests.)

Medians and 95% CIs are shown on the left side of the figure. The pattern shown here is different from the one in Fig 6, which shows that age affected only the intercept of exponentials fit to the NCPT GI. There are differences between pairs of subtests on each of the 3 parameters. The right side of the figure shows the level of significance of contrasts between each pair of subtests on each exponential parameter. The statistical tests are based on the difference scores between pairs of subtests on individual bootstrap iterations. The Benjamini-Hochberg correction (False Discovery Rate, FDR) [33] was used to control for multiple comparisons (the 21 pairwise subtest comparisons for a given exponential parameter).

Some salient differences between subtests are evident. The maximum amount of transfer from CT (Asymptote—Intercept) is greatest for AR, the only subtest involving numerical problem solving. The benefit of repeated testing (Intercept) is large for DSC, TMA, and TMB, the three subtests that emphasize speed of processing. The high intercept and low rate for TMA may provide a further dissociation (along with age) between direct practice and transfer. Some absences of difference are also noteworthy. The maximum amount and rate of transfer to FVMS and RVMS, which engage similar types of visual memory, are not significantly different. Similarly, PM, which assesses a very general ability that might be engaged by other subtests, has values similar to many of the other subtests on the three parameters.

It is difficult to disentangle the extent to which parameter values of the subtests reflect properties of cognition they assess, content of the CT program, or idiosyncratic features of the subtests themselves. But, whatever the cause, the pattern of differences indicates that all three parameters of D-R functions, not just the intercept, are subject to influence by factors. Absence of differences in rate and asymptote–intercept would therefore appear to be specific to age.

**Table 5. Intercept, rate, and asymptote–intercept of fit exponential functions for NCPT subtests.**

| NCPT Subtest | Cognitive Function(s) Assessed | Intercept (subtest points) | Rate (game$^{-1}$) | Asymp—Int (subtest points) |
|---|---|---|---|---|
| Arithmetic Reasoning | Numerical Problem Solving | 0.4214 | 0.00559 | 3.6207 |
| Digit Symbol Coding | Speed of Processing & Memory | 1.0587 | 0.00445 | 2.5728 |
| Forward Visual Memory Span | Visual Short-Term Memory | 0.9810 | 0.00581 | 2.5262 |
| Reverse Visual Memory Span | Visual Working Memory | 0.4369 | 0.00729 | 2.1506 |
| Trail Making A | Speed of Processing | 1.3657 | 0.00266 | 2.5057 |
| Trail Making B | Speed of Processing & Cognitive Flexibility | 1.0274 | 0.00650 | 1.9667 |
| Progressive Matrices | Problem Solving & Fluid Reasoning | 0.5450 | 0.00452 | 2.5441 |

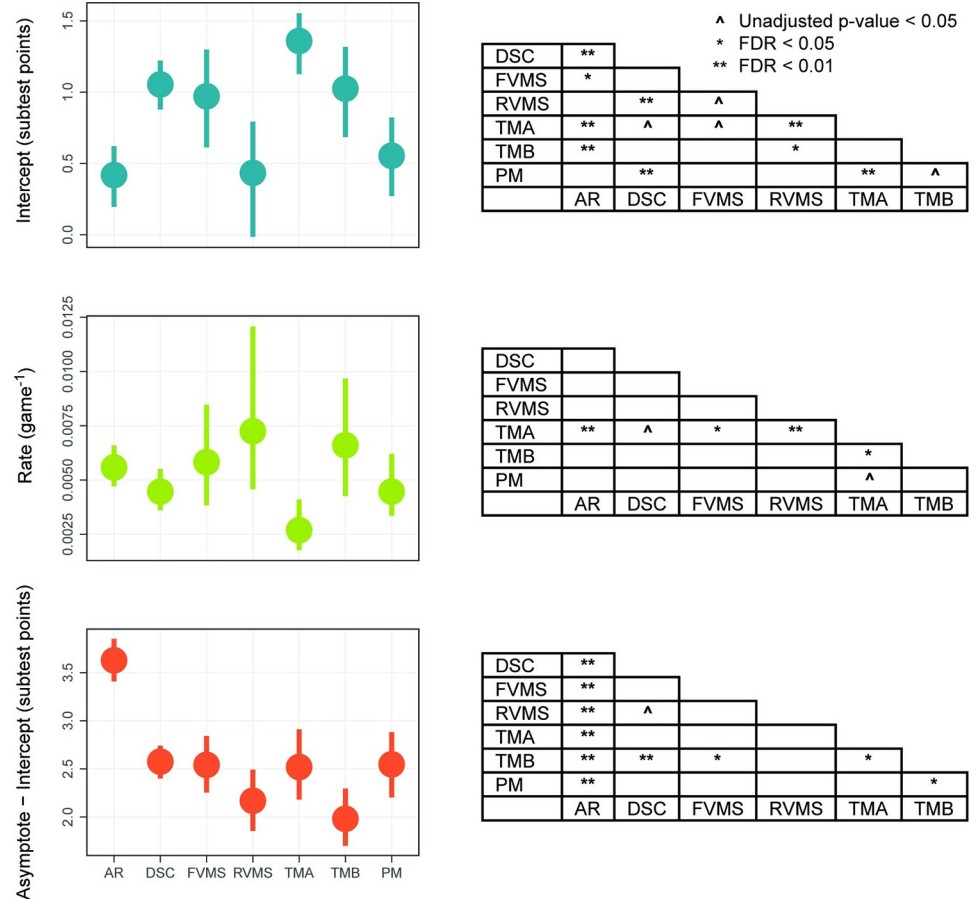

**Fig 9. Medians, 95% CIs, and comparisons between parameter estimates for individual NCPT subtests.** Left: medians (center points) and 95% CIs (upper and lower edges) for intercept, rate, and asymptote-intercept of exponential functions fit to each subtest. Right: statistical significance for each comparison of parameter estimates between subtests. Medians, CIs, and p-values were determined through a bootstrapping procedure. The Benjamini-Hochberg correction (False Discovery Rate, FDR) was used to control for multiple comparisons: ** = FDR < 0.01, * = FDR < 0.05, ^ = uncorrected p < 0.05, blank = uncorrected p > 0.05.

## Age differences on individual subtests of the NCPT

Earlier, D-R functions for GI were shown to differ between age groups. These differences appeared to result from the effects of repeated testing (indexed by the intercept), but not from transfer of CT (indexed by the asymptote–intercept and rate parameters). The GI, however, is an overall measure of performance based on the combined scores of the individual subtests (Methods). The last section reported evidence that the subtests varied in their responsiveness to both repeated testing and transfer. If age differences in transfer did occur on any of the individual subtests, they may have been obscured in the GI. This section therefore examines age differences in the D-R functions for each individual NCPT subtest.

Fig 10 shows how each D-R parameter correlated with age for each scaled subtest. These parameters were estimated by applying the model used earlier for age (Model 2, Methods) to each subtest separately. The 95% CIs and p-values are for Pearson correlation coefficients and based on replications from 1000 bootstrap iterations. Resampling was done with replacement and preserved the number of participants at each age level. On each iteration, the D-R parameters were estimated and correlated with the four levels of age. P-values are for two-tailed tests

### Intercept (GI points)

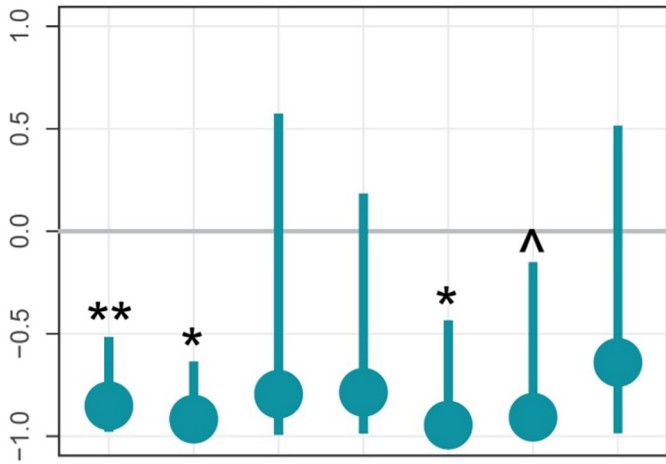

### Rate (game⁻¹)

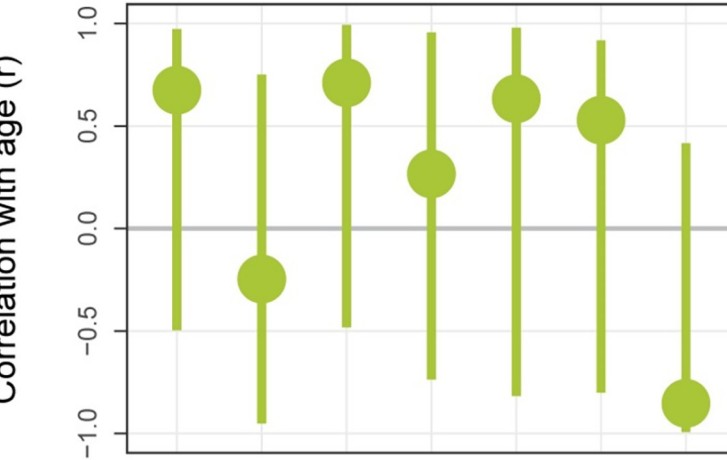

### Asymptote − Intercept (GI points)

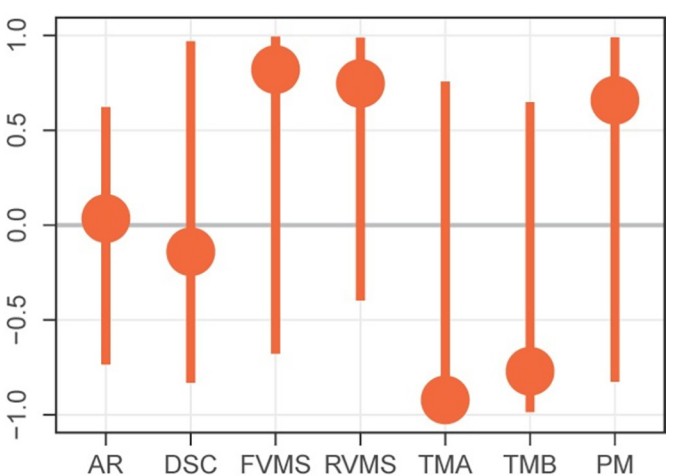

**Fig 10. Medians, 95% CIs, and statistical significance of correlations between age level and each exponential parameter for each NCPT subtest.** The medians (center points), 95% CIs (upper and lower edges), and p-values of the correlations were determined through a bootstrapping procedure. The Benjamini-Hochberg correction (False Discovery Rate, FDR) was used to control for multiple tests: ** = FDR < 0.01, * = FDR < 0.05, ^ = uncorrected p < 0.05, blank = uncorrected p > 0.05.

of whether a correlation could be zero and adjusted for multiple tests (3 parameters x 7 subtests = 21 tests) using the B-H (FDR) method.

The results for the individual subtests are consistent with those found for GI. The correlation between age and intercept is numerically negative for all subtests and significantly so for four. The correlations between age and the other two parameters are numerically negative for some subtests and positive for others. None are statistically significant. So it would be expected that, when the individual subtest scores are combined to form GI, age differences in asymptote–intercept and rate would not be evident.

The pattern of correlations involving the intercept provides clues about why the effect of repeated testing diminished with age. Performance on the four subtests that correlate significantly with age depends more on speed than performance on the other three. Briefly (see Methods and S2 Appendix for more details), the dependent measure on TMA and TMB is speed, and responding quickly on AR and DSC enables more (potentially correct) responses to be made within a limited amount of time. In contrast, good performance on FVMS, RVMS, and PM depends only on accuracy (which may be reduced by responding quickly). Perhaps an emphasis on speed helped younger participants to benefit from repeated testing more than older ones. Younger participants might learn to go faster with greater ease because they have faster cognitive processes or because older individuals adopt a more conservative tradeoff between speed and accuracy [34].

## Discussion

Here we used the power afforded by a large-scale dataset to examine the effects of dose on transfer from CT. D-R functions relating amount of training on an online CT program to the amount of improvement on a neuropsychological test battery were obtained for approximately 107,000 individuals. Separate D-R functions were obtained for different levels of several demographic factors and each subtest of the battery. Besides providing information about the precise form of the D-R relationship and its generality, the results of this study bear on the efficacy of CT and how it changes across the lifespan. They also provide intriguing clues about basic mechanisms of learning and cognitive aging, as well as illustrate some methodological affordances of D-R functions. These substantive and methodological implications are discussed further below.

### The generality and significance of dose effects on CT

The dose-response relation for CT was examined on each of several tests assessing different cognitive faculties and across demographic variables, all within a single study involving a sufficient number of participants to provide a high level of sensitivity. In contrast to meta-analyses, which combine data across often disparate studies, a consistent relation was found between CT dose and the amount of transfer. Monotonically increasing D-R functions, well fit by an exponential approach to an asymptote, were found to be the general case: Increasing amounts of CT resulted in greater transfer to overall NCPT performance, performance on the individual subtests, and at each level of age, education, and gender. This consistent relation would suggest that low dosage may be responsible for some studies failing to find effects of CT, especially large ones with high sensitivity (e.g., Owen et al. [16]).

It should be noted, however, that the results of our study differ dramatically from those of a recent online study by Stojanoski et al. [35] that also examined D-R relations between CT and performance on a neuropsychological test battery. Participants in that study were asked if they engaged in CT and, if so, for how long. Among the approximately 1000 individuals who answered affirmatively, no relation was found between their self-reported duration and performance on the test battery. Moreover, no differences in test performance were found between these participants and approximately 7,000 who reported having not engaged in CT. The authors interpreted these results as evidence that CT does not improve general cognitive abilities.

As mentioned, CT effects observed in RCTs tend to be modest. Differences in the results of the two studies might therefore be due to methodological differences related to sensitivity. First, the sample of individuals who engaged in CT was approximately 100 times larger in the present study. Second, our measures of test performance controlled for each participant's baseline level prior to CT: To examine changes from baseline performance, participants were tested before and after CT. In the Stojanoski et al. [35] study, participants were tested once (after either having engaged in CT or not). Third, we recorded the actual number of gameplays. Stojanoski et al. inferred the amount of CT from a less direct measure: the length of time participants reported having engaged in CT.

The present study also examined the influence of different factors on individual parameters of D-R functions. This enabled us to measure separately how these factors modulated effects on the NCPT of transfer from CT vs. direct practice from repeated testing. Both types of effect were found to vary across subtests. Possible sources for this variation include differences in the malleability of the assessed cognitive faculties, the particular types of training provided by the CT program, and/or features specific to the subtests themselves. Future research could untangle the contributions of these sources by assessing performance with other cognitive tests or comparing participants who trained with different sets of games. In contrast to subtest, we did not detect effects of either gender or education on the change in NCPT performance following CT. Note, however, that Guerra-Carillo et al. [36] did find greater education to be associated with greater change on the NCPT following training with the same CT program as in the present study. Given their analyses, it cannot be determined whether greater improvement on the NCPT was due to more transfer from CT or to a larger effect of direct practice. Either effect, however, would indicate that education can improve subsequent learning.

## The efficacy of CT across the adult lifespan

Among the most intriguing findings of the present study are differences in the D-R functions across age groups. While the effects of direct practice on the NCPT diminished with age, those of transfer from CT remained constant. This pattern of effects was found both for overall NCPT and some of the individual subtests. Greater modulation of direct learning by age was found for subtests that emphasized speed. Perhaps an emphasis on speed helped younger participants to benefit from repeated testing more than older ones. The slowing of reaction times with age is well documented [37], though it remains unclear to what extent it is due to the slowing of cognitive processes vs. adopting a more conservative strategy for trading between speed and accuracy [34]. But either underlying cause might enable younger participants to improve their speed on these subtests with greater ease.

Especially interesting is the absence of age differences in transfer, which replicates and extends the findings of an RCT involving the same CT program and assessments. In RCTs of CT, both the treatment and control groups involve repeated testing, and are thus equated for direct learning of the assessments. So the treatment effects (difference between groups in their

respective changes on assessments) should include only transfer. Our findings therefore imply that an RCT of CT should find equivalent treatment effects in the old and young. Such findings were indeed observed in Ng et al.'s [19] re-analysis of Hardy et al.'s [18] RCT. No significant differences were found between the effects of CT on a younger (18–49) and older group (50–80) of participants. In the present study, no diminution in the efficacy of CT was found for a yet older group (71–89) of participants.

The absence of significant age differences in the effects of CT may seem surprising given the neural and cognitive changes known to accompany normal aging (Park & Reuter-Lorenz [38]). These include changes across much of the adult lifespan in the effects of direct practice on a wide variety of activities, including both the NCPT and Lumosity games (e.g., Sternberg et al. [39]). Though age modulates the effects of direct practice on the NCPT and Lumosity games, this is not the case for transfer of learning from the games to the NCPT. This suggests that there may be differences between direct practice and transfer in what is learned and/or how learning occurs. Moreover, the differences between these two forms of learning may be related to which cognitive abilities are preserved and which change over the course of normal adult aging.

The present findings suggest that direct practice involves learning that diminishes with cognitive aging while transfer involves only (or mostly) learning that does not. What types of learning are diminished vs. preserved across the adult lifespan has yet to be fully elucidated. There is, however, a broad consensus that the type of learning that can be expressed verbally and whose sources can be consciously recollected (declarative episodic memory) diminishes in old age, while the type of nonverbal procedural learning that underlies skills is preserved [40]. While both types of learning might mediate improved performance from direct practice, transfer might involve only the latter. In CT, procedural learning might lead to more effective use of cognitive skills that underlie performance during both training and assessment, such as those involving attention, working memory, or executive processes. Whatever they ultimately may be found to be, the differences between direct practice and transfer intimated by our findings have broad implications. Besides CT and cognitive aging, they bear also on basic questions about mechanisms of learning and memory, e.g., the number and types of cognitive and neural systems involved [41].

## Methodological affordances of D-R functions

The D-R functions enabled by large datasets have a variety of methodological affordances. An affordance illustrated in the present study is the ability to measure how their individual parameters are modulated by various factors, such as demographic ones or to which cognitive faculty an assessment is sensitive. Such measurements enable inferences about mechanism, here the mechanisms of cognitive aging and learning through transfer and direct practice. For example, as discussed above, we found evidence that, unlike direct practice, transfer may consist only (or largely) of the type(s) of learning that are preserved across the lifespan. Another affordance may be relevant to clinical research and practice. Given a consensus about how big a change is meaningful on a particular neuropsychological assessment, pre-existing D-R functions for that assessment could help to determine 1) whether CT could produce an effect of that magnitude in a particular patient group and 2) how much training would be necessary.

D-R functions for CT may also provide a useful converging operation to control groups. Each controls in a somewhat different way for changes on assessments due to causes unrelated to the treatment. While use of a control group to measure transfer involves the comparison of change on an assessment between different training conditions, D-R functions compare the changes on an assessment following different amounts of the same type of training. Like any

control group, D-R functions enable measurement of transfer free from confounding effects of repeated testing. And, like active control groups, they help to minimize nonspecific effects of motivation and/or expectations that might arise from the mere fact of engaging in any form of training (but see Limitations below). D-R functions, however, also circumvent an unintended consequence of training provided by active control groups. Such training can sometimes result in real cognitive benefits and thus reduce the apparent effects of CT (treatment–control). An example is training on crossword puzzles. Its effects on an assessment, albeit less than those of the CT treatment, can be seen in Fig 4 of Hardy et al. [18]. Note that, without the D-R functions displayed in the figure, effects of transfer in the control group would not have been detectable in this RCT.

## Limitations

The present study has a number of limitations. Some stem from its observational and cross-sectional design. Each dose level in the same D-R function involved a different set of participants, and these participants were not randomly assigned to that level. Rather, each participant decided upon their own dose, i.e., the amount of CT they engaged in between the two NCPT assessments. This leaves open the possibility that participants with different doses differed in other ways that influenced the amount of change on the NCPT. Despite that, the positive relation between dose and response cannot be explained by demographic differences between individuals with more or less training, as this relation occurred at each level of age, education, and gender. Other individual differences that might have both influenced change on the NCPT and been correlated with the amount of CT include motivation and expectation of improvement [9]. While it cannot be conclusively rejected, an explanation attributing all positive D-R relations found in the present study to motivation or expectations would be far from parsimonious. Such an explanation would need to account for both the differences between subtests and lack of differences between demographic groups in the amount that change on the NCPT increased with dose. Concerns about motivation or expectations in future studies with designs like ours might be addressed through self-reports or questionnaires.

Other limitations stem from the broad range of CT, which involved a large number of different games and targeted multiple cognitive domains. Participants were free to choose which games to engage in, and daily suggestions were offered to encourage diversity (see Methods). While such cross-training may be an effective form of CT, it made it difficult to study the effects of training in specific cognitive domains. Nonetheless, future analyses of this dataset could classify participants with respect to the relative amounts of training they received in each of several cognitive domains. As mentioned, the games can be organized on the basis of their primary cognitive demands (Memory, Attention, Flexibility, Problem Solving, Speed of Processing, Math Skills, or Language Skills). Alternatively, each game can be characterized by its loadings on a set of latent cognitive factors [42]. A more direct approach, involving other studies, would be to limit CT to a specific cognitive domain. While there exist many studies of targeted CT, a further requirement for the types of analyses performed here is a dataset large enough to obtain precise D-R functions. Perhaps the widespread application of digital therapeutics involving targeted CT to treat specific neuropsychological disorders [43] will provide such datasets in the future.

## Future directions

One direct extension of the present work would be to examine the effects on D-R functions of additional factors. Such factors might include neuropsychological disorders, e.g., ADHD or MCI. Examining the extent to which they influence transfer and direct learning might provide

further information about their cognitive dimensions, as well as shed additional light on the basic mechanisms of transfer and why it might be preserved in advanced age. Other factors might include variations in CT content (e.g., cognitive domain, emphasis on speed vs. accuracy, type of feedback) or delivery (e.g., spaced vs. massed practice, amount of social contact, guidance) with the aim of improving efficacy. By enabling the rate and maximum amount of transfer to be measured, D-R functions provide a more complete picture of how a factor influences the efficacy of CT. Moreover, if a factor affects the shape of the function, conclusions based on one dose of CT may not generalize to others.

A less direct but promising extension could involve more detailed models of how transfer from CT and direct learning from repeated testing combine to influence performance on an assessment. For example, Steyvers et al. [44] proposed a model in which both types of learning are described by exponential functions and combine additively. The model examines the dynamics of both over the course of multiple successive assessments delivered during CT to the same individuals. Importantly, this type of longitudinal data reduces potential correlations between amount of CT and individual differences that might influence performance on assessments (see Limitations). The model also incorporates CT prior to the first assessment, allowing inclusion of additional participants (see Preliminary analyses in Results).

An issue of especial relevance to CT that might be addressed through modeling concerns forgetting. Perhaps separate processes of learning and forgetting could be modeled together. Like learning, forgetting could involve either the results of direct practice on assessments or transfer from CT. Both types of forgetting might be expected to occur in any study evaluating CT and to depend on individual differences (e.g., age or neuropsychological status) and task factors (e.g., cognitive domains of CT and assessments). To obtain different amounts of forgetting in a study, the time intervals between assessments and between training sessions could be varied. Each type of forgetting would be represented as a variable (e.g., rate of exponential decay) in a model fit to the study results. The persistence over time of transfer from any given program of CT ought to depend on, and might be predicable from, the rate at which it is forgotten.

## Conclusions

The present study illustrates how large datasets can enable precise measurements of D-R functions for CT. These functions were found to be present and well approximated by an exponential approach to an asymptote across demographic factors and neuropsychological subtests. Their robust presence and consistent form suggest that low dosage may be responsible for at least some studies failing to find effects of CT. Also illustrated is how D-R functions can assess separately the effects of both direct practice and transfer. Especially interesting is that, while the effects of direct practice on the NCPT diminished with age, those of transfer from CT remained constant. Besides offering good news to older individuals practicing CT, this finding suggests that direct practice and transfer do not involve identical learning processes, with the latter being limited to learning processes that remain constant across the adult lifespan. In conclusion, the results of this study provide a further demonstration of the practical and theoretical benefits of an approach to CT that seeks to identify the conditions on which it depends and to characterize their relations. Among the benefits, we believe, will be greater progress towards better defining and clearly answering questions about whether CT "works."

## Supporting information

**S1 Appendix. Descriptions of the CT games used in the present study.**
(PDF)

**S2 Appendix. Subtests from the NeuroCognitive Performance Test (NCPT) used in the present study.**
(PDF)

**S1 File. D-R function of effect sizes for NCPT Grand Index.**
(PDF)

**S2 File. Age as a continuous variable.**
(PDF)

## Acknowledgments

We thank Murali Doraiswami, Ben Katz, Kevin Madore, and Mark Steyvers for their helpful comments on drafts of this article.

## Author Contributions

**Conceptualization:** Allen M. Osman, Nicole F. Ng, Robert J. Schafer.

**Data curation:** Paul I. Jaffe.

**Formal analysis:** Allen M. Osman, Paul I. Jaffe, Robert J. Schafer.

**Methodology:** Allen M. Osman, Robert J. Schafer.

**Project administration:** Kelsey R. Kerlan.

**Writing – original draft:** Allen M. Osman, Paul I. Jaffe, Robert J. Schafer.

**Writing – review & editing:** Allen M. Osman, Paul I. Jaffe, Nicole F. Ng, Kelsey R. Kerlan, Robert J. Schafer.

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
