## [Decision Letter · Decision Letter 0]

15 Jul 2022

PONE-D-22-16722Transfer of learning: Analysis of dose-response functions from a large-scale, online, cognitive training datasetPLOS ONE

Dear Dr. Osman,

Thank you for submitting your manuscript to PLOS ONE. After careful consideration, we feel that it has merit but does not fully meet PLOS ONE’s publication criteria as it currently stands. Therefore, we invite you to submit a revised version of the manuscript that addresses the points raised during the review process.

We look forward to receiving your revised manuscript.

Kind regards,

Shadab Alam, Ph.D.

Academic Editor

PLOS ONE

Journal Requirements:

2.Please provide additional details regarding participant consent. In the ethics statement in the Methods and online submission information, please ensure that you have specified what type you obtained (for instance, written or verbal, and if verbal, how it was documented and witnessed). If your study included minors, state whether you obtained consent from parents or guardians. If the need for consent was waived by the ethics committee, please include this information.

"I have read the journal's policy and the authors of this manuscript have the following competing interests: The present study examined effects of cognitive training with Lumosity on the NeuroCognitive Performance Test, both of which are produced by Lumos Labs, Inc.  All authors are current paid employees of the company, and all hold stock in the company."

4. Thank you for providing the following Funding Statement:  

"No external funding contributed to this research; Lumos Labs, Inc. supported the research through the development of its software tools and employment of the authors. The specific roles of these authors are listed in the Author Contributions section. Lumos Labs had no other role in the study design, data collection and analysis, decision to publish, or preparation of the manuscript. Legal approval for publication before submission of the manuscript was obtained from Lumos Labs."

We note that one or more of the authors is affiliated with the funding organization, indicating the funder may have had some role in the design, data collection, analysis or preparation of your manuscript for publication; in other words, the funder played an indirect role through the participation of the co-authors. 

If the funding organization did not play a role in the study design, data collection and analysis, decision to publish, or preparation of the manuscript and only provided financial support in the form of authors' salaries and/or research materials, please review your statements relating to the author contributions, and ensure you have specifically and accurately indicated the role(s) that these authors had in your study in the Author Contributions section of the online submission form. Please make any necessary amendments directly within this section of the online submission form.  Please also update your Funding Statement to include the following statement: “The funder provided support in the form of salaries for authors [insert relevant initials], but did not have any additional role in the study design, data collection and analysis, decision to publish, or preparation of the manuscript. The specific roles of these authors are articulated in the ‘author contributions’ section.” 

If the funding organization did have an additional role, please state and explain that role within your Funding Statement. 

Please also provide an updated Competing Interests Statement declaring this commercial affiliation along with any other relevant declarations relating to employment, consultancy, patents, products in development, or marketed products, etc.  

Additional Editor Comments:

Dear author(s)

Thank you for your submission to the journal. Overall the paper is good but based on the reviewer's comments, your manuscript needs minor corrections before being accepted for publication. Kindly make appropriate changes to the manuscript and submit the revised copy of the manuscript based on the reviewer's comments.

Reviewers' comments:

Reviewer's Responses to Questions

**Comments to the Author**

1. Is the manuscript technically sound, and do the data support the conclusions?

Reviewer #1: Yes

Reviewer #2: Yes

2. Has the statistical analysis been performed appropriately and rigorously? 

Reviewer #1: Yes

Reviewer #2: Yes

3. Have the authors made all data underlying the findings in their manuscript fully available?

Reviewer #1: Yes

Reviewer #2: Yes

4. Is the manuscript presented in an intelligible fashion and written in standard English?

Reviewer #1: Yes

Reviewer #2: Yes

5. Review Comments to the Author

Reviewer #1: Transfer of Learning: Analysis of Dose-Response Functions from a Large-Scale, Online, Cognitive Training Abstract

I appreciate the opportunity to review this article. Being immersed in this literature with Lampit et al., I read this with much curiosity. Comments and suggestions are below.

1. Overall, this is well written, but not necessarily easy to follow. This is very dense material which is not necessarily a bad thing, but being someone who is very familiar with the literature, it was difficult to understand.

2. Methodologically, this is a strong article with a large sample size.

3. I am not sure I understand what DR functions are after reading this article (lines 58-60). I guess it has something to do with background characteristics and such. I was never quite sure what this meant. To me, this is simply looking at covariates of dose-response.

4. The review and integration of the literature is on point.

5. I think GI stand for Grant Index (line 229). I figured this out later. So I assume this is the just the overall global cognitive composite.

6. I was glad to see Figure 5 which shows that after a while, there is diminishing return on the therapeutic benefits of these games; that parallels Lampit et al.’s work. It is not clear to me how much that translates into time, it is only express in games. As someone who works with these games, I need to know TIME for this to be useful.

7. I had trouble interpreting Figure 3, but the other figures made sense (more or less)

8. A strength of this study is that they examined the impacts of training on various cognitive domains (Table 5 and Figure 8). What is not there is what type of training is provided; it seems to be all lumped together (and I get it…there is only so much you can do with the data).

9. Some of the figures were blurry and hard to read. I just really gave up and tried to glean what I could from the text.

10. Some interesting findings

a. “we did not detect effects of either gender or education on the change in NCPT performance following CCT”

b. “While the effects of direct practice on the NCPT diminished with age, those of transfer from CT remained constant”

11. Line 633 – Not sure what is meant by “a useful converging operation”.

12. In the Limitation section, one could speculate on how personality could also impact the cognitive training outcomes. There has been some recent work on that.

13. In the limitations, I am glad to see that “motivation” and “expectation of improvement” is provided. These are important components.

14. Overall, I think this article will be seminal, but it is very obtuse and difficult to read because it is very dense. I defer to other reviewers in someone making this more readable if possible.

Reviewer #2: The review is as an attachment:

Overall a very cool study and well done. I think by making the results section clearer this will turn out to be a great paper. It would be cool to see future research at Luminosity minimizing confounders by altering the number of prompts/notifications or changes in the app followed by a regression-discontinuity design.

In the interest of transparency, I sign all my reviews the comments made are meant to be constructive and I think that this paper has great promise .

Nicholas Judd, Ph.D.

6. PLOS authors have the option to publish the peer review history of their article (what does this mean?). If published, this will include your full peer review and any attached files.

Reviewer #1: No

Reviewer #2: **Yes: **Nicholas Judd

---

## [Author Response · Author response to Decision Letter 0]

19 Sep 2022

Reviewer 1

I appreciate the opportunity to review this article. Being immersed in this literature with Lampit et al., I read this with much curiosity. Comments and suggestions are below.

1. Overall, this is well written, but not necessarily easy to follow. This is very dense material which is not necessarily a bad thing, but being someone who is very familiar with the literature, it was difficult to understand.

We have responded to a number of your comments below (3, 5, 6, 7, 9, 11) in ways that we think will make the paper somewhat easier to follow. Thanks especially for comment 6, which helped us make the paper meaningful to a larger audience.

2. Methodologically, this is a strong article with a large sample size.

Thanks. We think that the large sample size is one of the study’s greatest strengths.

3. I am not sure I understand what DR functions are after reading this article (lines 58-60). I guess it has something to do with background characteristics and such. I was never quite sure what this meant. To me, this is simply looking at covariates of dose-response.

The concept of a dose-response (D-R) function is central to the paper. After reading your comment, we realized that we never provided an explicit definition. One can now be found on line 63.

4. The review and integration of the literature is on point.

Thanks. We are encouraged by your feedback.

5. I think GI stand for Grant Index (line 229). I figured this out later. So I assume this is the just the overall global cognitive composite.

That’s correct. It’s important that we be clear about this, since GI refers to a major dependent variable of the study. After reading your comment, we realized that, when we defined the Grand Index (on what was formerly line 229), we never indicated that its abbreviation was GI. This has been rectified (line 322). We also provide a reminder on line 482 that GI is a measure of overall performance on the NCPT. 

6. I was glad to see Figure 5 which shows that after a while, there is diminishing return on the therapeutic benefits of these games; that parallels Lampit et al.’s work. It is not clear to me how much that translates into time, it is only express in games. As someone who works with these games, I need to know TIME for this to be useful.

We agree that knowing dose in terms of the amount of time spent training is important for clinical research and practice. Since, a single game lasts on average about 3 minutes, the cumulative number of hours spent training is approximately equal to the number of games played divided by 20. This is now stated where the to-be-interpreted features of the D-R plot are introduced (line 481). 

People in clinical fields might also be interested in S1 Additional Analyses, where effect size (Hedge’s g) is plotted as a function of training (both number of games and time). 

7. I had trouble interpreting Figure 3, but the other figures made sense (more or less)

To help make Fig. 3 more interpretable, the figure caption has been revised somewhat (lines 445-448).

8. A strength of this study is that they examined the impacts of training on various cognitive domains (Table 5 and Figure 8).

Agreed.

8. Cont. What is not there is what type of training is provided; it seems to be all lumped together (and I get it…there is only so much you can do with the data).

We agree and elaborate on this point in the Limitations section (lines 1035-1044).

 “Other limitations stem from the broad range of CT, which involved a large number of different games and targeted multiple cognitive domains. Participants were free to choose which games to engage in, and daily suggestions were offered to encourage diversity (see Methods). While such cross-training may be an effective form of CT, it made it difficult to study the effects of training in specific cognitive domains. Nonetheless, future analyses of this dataset could classify participants with respect to the relative amounts of training they received in each of several cognitive domains. As mentioned, the games can be organized on the basis of their primary cognitive demands (Memory, Attention, Flexibility, Problem Solving, Speed of Processing, Math Skills , or Language Skills). Alternatively, each game can be characterized by its loadings on a set of latent cognitive factors [41]. A more direct approach, involving other studies, would be to limit CT to a specific cognitive domain. …”

9. Some of the figures were blurry and hard to read. I just really gave up and tried to glean what I could from the text.

We apologize for this. We had failed to detect that resolution in some of the figures was reduced during submission. The figures in the resubmission are all 300 dpi.

10. Some interesting findings

a. “we did not detect effects of either gender or education on the change in NCPT performance following CCT”

Still, we do accept the possibility (and, in fact, hope) that education can influence the effects of CT (lines 928-932).

b. “While the effects of direct practice on the NCPT diminished with age, those of transfer from CT remained constant”

We consider this to be one of the major findings of the study!

11. Line 633 – Not sure what is meant by “a useful converging operation”.

We have revised the sentence containing this phrase (lines 997-998) to clarify its meaning.

12. In the Limitation section, one could speculate on how personality could also impact the cognitive training outcomes. There has been some recent work on that.

This is definitely an interesting question and one directly related to the important goal of individualizing CT. We are very interested in traits like growth mindset and have informally observed a variety of ways in which individual participants engage with our CT program. But we are not yet able to make public statements, even speculations, on this topic.

13. In the limitations, I am glad to see that “motivation” and “expectation of improvement” is provided. These are important components.

We agree.

14. Overall, I think this article will be seminal, but it is very obtuse and difficult to read because it is very dense. I defer to other reviewers in someone making this more readable if possible.

We hope you’re right about the article’s impact. Thanks for helping us make it more accessible and of interest to a wider audience. 

Reviewer 2

Overall I really enjoyed reading this paper – the methods were well done and of high quality yet they need to be clarified to make this manuscript suitable for publication. In the future, it would be good to see something preregistered or with a held-out test set. The data is openly accessible which is great, while I think code should be included it’s not a requirement. 

1. The intro is well written and I have no comments except I think the use of CT & CCT is unnecessary (along with GI, which after being introduced could just be called ‘improvement or change’; and cognitive training can just be written as training after you introduce it) yet this is the author's stylistic choice and is a mere suggestion. 

While we’re going to stick with CT and GI, CCT has been replaced with just CT.

The main section that I think needs to be re-written for clarity is the results section; this may also be due to the methods section lacking the relevant info. I have read it a few times and it is still unclear this can be partially fixed by making the tables and figures clearer – yet exactly what is happening needs to be explicitly explained. Simply things like reiterating that you are fitting a DR model to each subgroup at each section on the results (we fit X models per subgroup). 

You are absolutely right. An important shortcoming of the paper was the lack of clarity about the models used to obtain parameter estimates for the exponential functions fit to the data. While these models have not changed, they are now described in considerable detail (lines 331 – 411) in a new section in the Methods. It is also made clear in both the Methods and Results which models were applied to which analyses.

We think that, by making clear the actual models employed, these descriptions address most of your major point (9, a-f). Providing clear detailed descriptions of the models also strengthened the paper. Thanks for your feedback on this!

Specific points: 

2. Plots still need to be in higher resolution

Done. All figures are now 300 dpi. We again apologize for some of the original figures being too fuzzy to read.

3. Plots please also change the unadjusted significance to an (x) or another symbol other than a star, as an alpha of .05 should correspond to a single *.

Done. See Figs 9 and 10.

4. Plots 6, 9 & 10 are way too close to box plots with IQR, please change them to standard coefficient plots (you can revert the axis if you want the vertical). There are also dots on them and they are missing parts – yet if you make new coefficient plots this doesn’t matter. 

Done. They are now coefficient plots.

5. I would like standardized effects (mean = 0, sd = 1) throughout, the authors don’t make a strong claim about intelligence and IQ point so why scale by them... It's much clearer to a wide readership to be in standard units. Or at least standard units next to IQ points (for example line 298). 

We appreciate the comment, and the goal of clarity for a wide readership. However, the units used in this paper follow the convention used for the NeuroCognitive Performance Test, as first described in Morrison et al. (2015). Since its introduction the NCPT has been used in large healthy and clinical populations (e.g., as part of the UCSF Brain Health Registry), always with the units used here. Among the studies to use the NCPT, the large RCT conducted by Hardy et al. (2015) that is described in this paper also used these units. To maintain consistency with the literature and facilitate a comparison across studies, we believe it is appropriate to keep the units as written.

6. Amount of gameplay (line 273) this distribution is very wide and low… Why do you have subjects that never played games…? This can’t possibly be a normal population; it would be good to see a robustness check with meaningful ranges. Considering those that did no games as test-retest neglects they’re not normal subjects. Unless it is a non-contact control? But it doesn’t seem this is the case… these are weird subjects that signed up for Lumiostiy and did no luminosity and came back weeks later for t2???

As with other program memberships -- both traditional (e.g., gymnasiums) and online (e.g., online courses or app subscriptions) -- there is a wide range of program use, including a number of people who do not use the program for which they signed up. Lumosity is no different, and these are indeed normal participants. An important point is that all participants received an email reminder when they were eligible to take their NCPT test at the second timepoint (T2); therefore, even those participants who had not engaged with their training were notified to return. As you imply, it may have been difficult to expect participants to return for T2 with no notification or reminder.

7. Clarify that you are making age brackets because it’s a modeling limitation? The brackets you choose make sense to me and I have no criticism about that but it will clarify to the reader why you went from a continuous measurement. Also, include counts for these brackets in table 1!!! The mean & SD of age are unnecessary since you don’t use them yet the graph is super nice. 

Age brackets are now provided in the text on lines 550-551. The rationale for bracketing age to make it a categorical variable can be seen in the descriptions of Models 2 and 3 (new material, lines 362-411).

8. Time is related to the number of games. There is obviously some error in when subjects took the 2nd test (some doing it later than others). Show this & maybe prove it is independent (or small) of demographics or add it as a covariate. Have you thought about this? Maybe I missed it? 

Astute observation. We had indeed considered what effect variability in when participants took the second NCPT might have on the observed D-R functions. Such an effect, if it occurred, would be that our estimates of a positive dose-response relation are in fact overly conservative. This is because longer time intervals between the first and second NCPTs provide, not only the opportunity to play more games in the interim, but also a greater opportunity to lose the benefits of having taken the first NCPT at the time of the second. In actuality, the positive correlation between number of gameplays and days between the two NCPTs was quite small (0.0069). For the above reasons, and because an adequate treatment would be rather lengthy, we had decided previously that a discussion of this issue was unnecessary and would be a distraction from the main narrative.

However, after reading your comment, we felt it important to include in the paper a discussion of this issue in relation to age effects on D-R functions. So, we included a new paragraph to show that the observed effect of age was not an artifact resulting from variability in time between the two NCPTs (lines 715 – 726). Briefly, we explain how the observed decrease in the effects of repeated testing (D-R function intercept) with age could result if there were a positive correlation between age and time interval between the two NCPTs. We then report the actual correlation, which was small and negative (-0.078). So, as with the positive relation between CT dose and amount of training, our estimate of age effects on the D-R functions is, if anything, overly conservative.

9. This is my major point: The results section corresponding to tables 3 & 4: It doesn’t help that you have table 2 which follows a standard table reporting format of linear models. These tables give a false impression everything is in a single model when in reality each line (row) corresponds to a single model – it doesn’t help for clarity that you subtract from the first model.

Tables 3 and 4 actually do report the results of a single model! The values reported in each table were obtained by fitting a single model containing several exponential functions, each governed by dummy variables corresponding to one of the involved demographic levels. For example, all values in Table 4 were estimated simultaneously by fitting a 12-parameter (3 exponential x 4 age levels) model.

The three models employed in this study are described in detail (lines 331-411) in a new section in the Methods. The descriptions include explicit equations for the models and indicate which analyses in the Results each was applied to. The models are also referenced in the Results.

We believe that comments a-f below are all based on the impression that each row in Tables 3 and 4 corresponded to a different model. While this was not the case, it does highlight the importance of providing explicit descriptions of the fit models, which we have now done.

a. This section and these tables need to become much clearer; spell out exactly what you are showing. (e.g., we fit # number of models, then we wanted to test the differences of X model from X model; The model for high school females ages 14- 40…) 

b. One way to make the tables clearer is not to subtract model 1 (i.e., the reference) from the other. 

c. It is unclear the statistical tests happening here, is it each model's coefficient from zero or is it the contract of betas between models? Some of your claims are on contrasts so if you would like to keep them you need to do a contrast test (preferably a full set; or at least the contracts for your claims age 1 vs age 2; age 2 vs age 3; age 3 vs age 4 etc…). If this isn’t the case the way the tables are made is suuuuper misleading. That is when a model fit on young people has a beta of .1 a statistical test is needed to determine if .08 in old people is significantly different. I couldn’t find this information clearly spelled out. 

d. Honestly, for the tables, I would literally write model 1; model 2; model 3 on each row and stop subtracting the reference model. I just realized you have an empty column where you could write “Model” above everything (and then get rid of the confusing reference-based subtraction; depending on how the statistical tests are done; point c).

e. Another clear way to show they are separate models is to write the number of subjects fit in each as a row of the table. 

f. It took me a while to figure out why the only age differences model (line 374) exists… I think this is due to the model framework used (i.e., necessitating small groups of covariates being fit)? What I am not a fan of is claiming because you didn’t an effect in the model before, these demographics – which you show at the start of the results to be related to gameplay – are 

unnecessary. This is not how covariate control works or should be used; a non sig covariate is still potentially doing something and it should be there based on causal a priori hypotheses (Wysocki, Lawson, Rhemtulla, 2022, 

Adv. in Methods and pract. in psyc. sci.). I sympathize with your justification for an ‘increase of power’ because you are fitting models to the unity of groups (e.g., 18-40, HS, F) yet it's not an adequate reason to ignore 

covariates. If you would like to keep it a reason please write and point out your smallest group as an example. It’s okay to write that you have modeling constraints and that you just can’t do it/are limited by DR functions (is this the main reason?). If you really, really want to focus on this analysis another way to proceed is to residualize the change score for those left-out demographics. Personally, I think this section should be put in the SI and referenced in a sentence in the analysis before. How you proceed with it is totally your choice – yet the justifications are what I have issues with.

g. I think lines 387 - 389 are a mistake, the table doesn’t seem to reflect this and has similar coefficients to Table 3? 

Those lines (plus the preceding one) state “Medians and 95% confidence intervals of the bootstrapped values at each age level are shown in the figure. Here the parameter values for non-reference levels are shown directly, rather than as differences from those at the reference level. As in Tables 3 and 4, the intercept can be seen to decrease with increasing age, while rate and asymptote–intercept remain relatively constant.”

We think you are saying that Table 4 shows parameter values similar to those in Table 3, but different from those in the figure (6). If so, you are correct that the parameter values in Table 4 agree with those for age in Table 3. (Which means the two analyses, each involving a different model, produced similar findings for age.) However, Table 4 and Figure 6 do in fact show the same values. Perhaps the apparent difference is due to how they are represented. In contrast to the two tables, in Figure 6 “the parameter values for non-reference levels are shown directly, rather than as differences from those at the reference level.”

The discussion is well written and I have very little to comment on except the following: 

10. Lines 623-632 could be entirely trimmed. 

We like this paragraph and want to emphasize the described affordances of D-R functions.

11. Line 633 has a very good point and should definitely stay. 

OK.

12. I have issues with the limitation section, I feel like with this type of study we should just own the limitation and not try to skirt around it or justify it. Major issue with lines 657-659; Grammatifal reasoning didn’t fit… you don’t even include it in the table… it can’t be used ‘when convenient’. If you have a really good reason you think this isn’t a limitation then spell it out, yet the current justifications don’t pass muster, and it's okay, every study has limitations that don’t need to be explained away.

While we think that discussing the parsimony of an alternative explanation for our findings is not unreasonable, we agree that our selective consideration of results from the grammatical reasoning subtest is somewhat arbitrary. So, we removed the sentence about grammatical reasoning that was on lines 657-659.

Overall a very cool study and well done. I think by making the results section clearer this will turn out to be a great paper. It would be cool to see future research at Luminosity minimizing confounders by altering the number of prompts/notifications or changes in the app followed by a regression-discontinuity design. 

Interesting idea. The size and nature of the Lumosity program could make it a good testing ground for measuring the impact of the frequency and type of test reminders or notifications. This could be done using regression-discontinuity, as mentioned, or a randomized, parallel "split test" between two product variants. A regression-discontinuity design might also be applicable elsewhere. 

In any case, thanks for helping us to improve the paper, especially its clarity with regard to the modeling.

---

## [Editor Report · Decision Letter 1]

27 Sep 2022

PONE-D-22-16722R1Transfer of learning: Analysis of dose-response functions from a large-scale, online, cognitive training datasetPLOS ONE

Dear Dr. Osman,

Thank you for submitting your manuscript to PLOS ONE. After careful consideration, we feel that it has merit but does not fully meet PLOS ONE’s publication criteria as it currently stands. Therefore, we invite you to submit a revised version of the manuscript that addresses the points raised during the review process.

ACADEMIC EDITOR: Thank you for submitting your manuscript to PLOS ONE. After careful consideration, we have decided that your manuscript can not be accepted in the current form and needs minor revision.

We look forward to receiving your revised manuscript.

Kind regards,

Shadab Alam, Ph.D.

Academic Editor

PLOS ONE

Journal Requirements:

Additional Editor Comments:

Thank you for submitting your manuscript to PLOS ONE. After careful consideration, we have decided that your manuscript can not be accepted in the current form and needs minor revision.
---

## [Author Response · Author response to Decision Letter 1]

7 Oct 2022

This is a response to the Editor.

Dear Authors,

My apology for this confusion and probably missing adding the comments that need your clarification and probably some minor modifications in the revised articles. You have responded to the following reviewer comments (comments 5, 6, and 7) that I am copying here in italics:

5. I think GI stand for Grant Index (line 229). I figured this out later. So I assume this is the just the overall global cognitive composite.

That’s correct. It’s important that we be clear about this, since GI refers to a major dependent variable of the study. After reading your comment, we realized that, when we defined the Grand Index (on what was formerly line 229), we never indicated that its abbreviation was GI. This has been rectified (line 322). We also provide a reminder on line 482 that GI is a measure of overall performance on the NCPT. 

6. I was glad to see Figure 5 which shows that after a while, there is diminishing return on the therapeutic benefits of these games; that parallels Lampit et al.’s work. It is not clear to me how much that translates into time, it is only express in games. As someone who works with these games, I need to know TIME for this to be useful.

We agree that knowing dose in terms of the amount of time spent training is important for clinical research and practice. Since, a single game lasts on average about 3 minutes, the cumulative number of hours spent training is approximately equal to the number of games played divided by 20. This is now stated where the to-be-interpreted features of the D-R plot are introduced (line 481).

People in clinical fields might also be interested in S1 Additional Analyses, where effect size (Hedge’s g) is plotted as a function of training (both number of games and time). 

7. I had trouble interpreting Figure 3, but the other figures made sense (more or less)

To help make Fig. 3 more interpretable, the figure caption has been revised somewhat (lines 445-448).

Editor: While reviewing the revised manuscript, I found that the line numbers mentioned by you do not reflect the alleged changes, especially for comments 5,6, and 7. You are requested to kindly see the comments and modify the manuscript accordingly. Further, the figures require a small description, which has been raised by reviewers in comments 6 and 7. Please check this part for figures 3 and 5 specifically, and overall for other figures also.

Thank you.

Dear Dr. Alam,

 We apologize for the miscited line numbers. It appears that all line numbers cited in our responses to both reviewers were wrong. This occurred because the format of the tracked revisions was changed when the marked-up manuscript was incorporated into the PDF by the Editorial Manager. The version that we prepared, and to which the cited line numbers refer, displayed deleted materials in balloons in the right margin. But, in the manuscript shown in the PDF, deleted materials are displayed within the text. This change in format produced a change in the total number of lines, causing all our cited line numbers to be off. Unfortunately, I did not detect the change in line numbers when I proofread the PDF prior to approving the submission.

 To provide the correct line numbers, we have included below a revised copy of our responses to all comments by both reviewers. The only changes made to these responses are the line numbers cited for revisions (highlighted in red). Everything else is the same as in the original responses. (We strongly recommend viewing these revisions in the attached file entitled Response to Editor, rather than in the relatively unformatted text inserted into boxes during resubmission.)

 You should now be able to find the revisions made previously in response to comments 5 – 7 by Reviewer 1. We believe that, once you have seen these revisions, you will view them positively. In particular, we hope you will agree that Figs 3 and 5 are already well described. An entire paragraph (lines 399-410) and much of a full section (lines 398-448) are devoted to describing Fig 5. The statement mentioned in our response to Comment 6 (relating total time training to number of games) is located prominently within that paragraph (lines 401-3). The heatmap shown in Fig 3 is described both in the text (lines 360-4) and figure caption (lines 371-4). The revisions we made to the figure caption should make the figure more comprehensible (compare the current content of the caption with the deleted text).

 Thank you very much for your editorial work on our manuscript. Again, sorry for the confusion.

Sincerely,

 Allen Osman,

 Corresponding author 

Reviewer 1

I appreciate the opportunity to review this article. Being immersed in this literature with Lampit et al., I read this with much curiosity. Comments and suggestions are below.

1. Overall, this is well written, but not necessarily easy to follow. This is very dense material which is not necessarily a bad thing, but being someone who is very familiar with the literature, it was difficult to understand.

We have responded to a number of your comments below (3, 5, 6, 7, 9, 11) in ways that we think will make the paper somewhat easier to follow. Thanks especially for comment 6, which helped us make the paper meaningful to a larger audience. 

2. Methodologically, this is a strong article with a large sample size.

Thanks. We think that the large sample size is one of the study’s greatest strengths.

3. I am not sure I understand what DR functions are after reading this article (lines 58-60). I guess it has something to do with background characteristics and such. I was never quite sure what this meant. To me, this is simply looking at covariates of dose-response.

The concept of a dose-response (D-R) function is central to the paper. After reading your comment, we realized that we never provided an explicit definition. One can now be found on lines 59-60.

4. The review and integration of the literature is on point.

Thanks. We are encouraged by your feedback.

5. I think GI stand for Grant Index (line 229). I figured this out later. So I assume this is the just the overall global cognitive composite.

That’s correct. It’s important that we be clear about this, since GI refers to a major dependent variable of the study. After reading your comment, we realized that, when we defined the Grand Index (on what was formerly line 229), we never indicated that its abbreviation was GI. This has been rectified (line 266). We also provide a reminder on line 401 that GI is a measure of overall performance on the NCPT.

6. I was glad to see Figure 5 which shows that after a while, there is diminishing return on the therapeutic benefits of these games; that parallels Lampit et al.’s work. It is not clear to me how much that translates into time, it is only express in games. As someone who works with these games, I need to know TIME for this to be useful.

We agree that knowing dose in terms of the amount of time spent training is important for clinical research and practice. Since, a single game lasts on average about 3 minutes, the cumulative number of hours spent training is approximately equal to the number of games played divided by 20. This is now stated where the to-be-interpreted features of the D-R plot are introduced (lines 401-3). 

People in clinical fields might also be interested in S1 Additional Analyses, where effect size (Hedge’s g) is plotted as a function of training (both number of games and time). 

7. I had trouble interpreting Figure 3, but the other figures made sense (more or less)

To help make Fig. 3 more interpretable, the figure caption has been revised somewhat (lines 371-4).

8. A strength of this study is that they examined the impacts of training on various cognitive domains (Table 5 and Figure 8). 

Agreed.

8 (cont.). What is not there is what type of training is provided; it seems to be all lumped together (and I get it…there is only so much you can do with the data).

We agree and elaborate on this point in the Limitations section (lines 814-823).

 “Other limitations stem from the broad range of CT, which involved a large number of different games and targeted multiple cognitive domains. Participants were free to choose which games to engage in, and daily suggestions were offered to encourage diversity (see Methods). While such cross-training may be an effective form of CT, it made it difficult to study the effects of training in specific cognitive domains. Nonetheless, future analyses of this dataset could classify participants with respect to the relative amounts of training they received in each of several cognitive domains. As mentioned, the games can be organized on the basis of their primary cognitive demands (Memory, Attention, Flexibility, Problem Solving, Speed of Processing, Math Skills , or Language Skills). Alternatively, each game can be characterized by its loadings on a set of latent cognitive factors [41]. A more direct approach, involving other studies, would be to limit CT to a specific cognitive domain. …”

9. Some of the figures were blurry and hard to read. I just really gave up and tried to glean what I could from the text.

We apologize for this. We had failed to detect that resolution in some of the figures was reduced during submission. The figures in the resubmission are all 300 dpi.

10. Some interesting findings

a. “we did not detect effects of either gender or education on the change in NCPT performance following CCT”

Still, we do accept the possibility (and, in fact, hope) that education can influence the effects of CT (lines 724-9).

b. “While the effects of direct practice on the NCPT diminished with age, those of transfer from CT remained constant”

We consider this to be one of the major findings of the study!

11. Line 633 – Not sure what is meant by “a useful converging operation”.

We have revised the sentence containing this phrase (lines 782-3) to clarify its meaning.

12. In the Limitation section, one could speculate on how personality could also impact the cognitive training outcomes. There has been some recent work on that.

This is definitely an interesting question and one directly related to the important goal of individualizing CT. We are very interested in traits like growth mindset and have informally observed a variety of ways in which individual participants engage with our CT program. But we are not yet able to make public statements, even speculations, on this topic. 

13. In the limitations, I am glad to see that “motivation” and “expectation of improvement” is provided. These are important components.

We agree.

14. Overall, I think this article will be seminal, but it is very obtuse and difficult to read because it is very dense. I defer to other reviewers in someone making this more readable if possible.

We hope you’re right about the article’s impact. Thanks for helping us make it more accessible and of interest to a wider audience. 

Reviewer 2

Overall I really enjoyed reading this paper – the methods were well done and of high quality yet they need to be clarified to make this manuscript suitable for publication. In the future, it would be good to see something preregistered or with a held-out test set. The data is openly accessible which is great, while I think code should be included it’s not a requirement. 

1. The intro is well written and I have no comments except I think the use of CT & CCT is unnecessary (along with GI, which after being introduced could just be called ‘improvement or change’; and cognitive training can just be written as training after you introduce it) yet this is the author's stylistic choice and is a mere suggestion. 

While we’re going to stick with CT and GI, CCT has been replaced with just CT.

The main section that I think needs to be re-written for clarity is the results section; this may also be due to the methods section lacking the relevant info. I have read it a few times and it is still unclear this can be partially fixed by making the tables and figures clearer – yet exactly what is happening needs to be explicitly explained. Simply things like reiterating that you are fitting a DR model to each subgroup at each section on the results (we fit X models per subgroup). 

You are absolutely right. An important shortcoming of the paper was the lack of clarity about the models used to obtain parameter estimates for the exponential functions fit to the data. While these models have not changed, they are now described in considerable detail (lines 275-338) in a new section in the Methods. It is also made clear in both the Methods and Results which models were applied to which analyses.

We think that, by making clear the actual models employed, these descriptions address most of your major point (9, a-f). Providing clear detailed descriptions of the models also strengthened the paper. Thanks for your feedback on this!

Specific points: 

2. Plots still need to be in higher resolution

Done. All figures are now 300 dpi. We again apologize for some of the original figures being too fuzzy to read.

3. Plots please also change the unadjusted significance to an (x) or another symbol other than a star, as an alpha of .05 should correspond to a single *. 

Done. See Figs 9 and 10.

4. Plots 6, 9 & 10 are way too close to box plots with IQR, please change them to standard coefficient plots (you can revert the axis if you want the vertical). There are also dots on them and they are missing parts – yet if you make new coefficient plots this doesn’t matter. 

Done. They are now coefficient plots.

5. I would like standardized effects (mean = 0, sd = 1) throughout, the authors don’t make a strong claim about intelligence and IQ point so why scale by them... It's much clearer to a wide readership to be in standard units. Or at least standard units next to IQ points (for example line 298). 

We appreciate the comment, and the goal of clarity for a wide readership. However, the units used in this paper follow the convention used for the NeuroCognitive Performance Test, as first described in Morrison et al. (2015). Since its introduction the NCPT has been used in large healthy and clinical populations (e.g., as part of the UCSF Brain Health Registry), always with the units used here. Among the studies to use the NCPT, the large RCT conducted by Hardy et al. (2015) that is described in this paper also used these units. To maintain consistency with the literature and facilitate a comparison across studies, we believe it is appropriate to keep the units as written.

6. Amount of gameplay (line 273) this distribution is very wide and low… Why do you have subjects that never played games…? This can’t possibly be a normal population; it would be good to see a robustness check with meaningful ranges. Considering those that did no games as test-retest neglects they’re not normal subjects. Unless it is a non-contact control? But it doesn’t seem this is the case… these are weird subjects that signed up for Lumiostiy and did no luminosity and came back weeks later for t2??? 

As with other program memberships -- both traditional (e.g., gymnasiums) and online (e.g., online courses or app subscriptions) -- there is a wide range of program use, including a number of people who do not use the program for which they signed up. Lumosity is no different, and these are indeed normal participants. An important point is that all participants received an email reminder when they were eligible to take their NCPT test at the second timepoint (T2); therefore, even those participants who had not engaged with their training were notified to return. As you imply, it may have been difficult to expect participants to return for T2 with no notification or reminder.

7. Clarify that you are making age brackets because it’s a modeling limitation? The brackets you choose make sense to me and I have no criticism about that but it will clarify to the reader why you went from a continuous measurement. Also, include counts for these brackets in table 1!!! The mean & SD of age are unnecessary since you don’t use them yet the graph is super nice. 

Age brackets are now provided in the text on lines 455-6. The rationale for bracketing age to make it a categorical variable can be seen in the descriptions of Models 2 and 3 (new material, lines 289-338). 

8. Time is related to the number of games. There is obviously some error in when subjects took the 2nd test (some doing it later than others). Show this & maybe prove it is independent (or small) of demographics or add it as a covariate. Have you thought about this? Maybe I missed it? 

Astute observation. We had indeed considered what effect variability in when participants took the second NCPT might have on the observed D-R functions. Such an effect, if it occurred, would be that our estimates of a positive dose-response relation are in fact overly conservative. This is because longer time intervals between the first and second NCPTs provide, not only the opportunity to play more games in the interim, but also a greater opportunity to lose the benefits of having taken the first NCPT at the time of the second. In actuality, the positive correlation between number of gameplays and days between the two NCPTs was quite small (0.0069). For the above reasons, and because an adequate treatment would be rather lengthy, we had decided previously that a discussion of this issue was unnecessary and would be a distraction from the main narrative.

However, after reading your comment, we felt it important to include in the paper a discussion of this issue in relation to age effects on D-R functions. So, we included a new paragraph to show that the observed effect of age was not an artifact resulting from variability in time between the two NCPTs (lines 549-557). Briefly, we explain how the observed decrease in the effects of repeated testing (D-R function intercept) with age could result if there were a positive correlation between age and time interval between the two NCPTs. We then report the actual correlation, which was small and negative (-0.078). So, as with the positive relation between CT dose and amount of training, our estimate of age effects on the D-R functions is, if anything, overly conservative.

9. This is my major point: The results section corresponding to tables 3 & 4: It doesn’t help that you have table 2 which follows a standard table reporting format of linear models. These tables give a false impression everything is in a single model when in reality each line (row) corresponds to a single model – it doesn’t help for clarity that you subtract from the first model. 

Tables 3 and 4 actually do report the results of a single model! The values reported in each table were obtained by fitting a single model containing several exponential functions, each governed by dummy variables corresponding to one of the involved demographic levels. For example, all values in Table 4 were estimated simultaneously by fitting a 12-parameter (3 exponential x 4 age levels) model.

The three models employed in this study are described in detail (lines 275-338) in a new section in the Methods. The descriptions include explicit equations for the models and indicate which analyses in the Results each was applied to. The models are also referenced in the Results.

We believe that comments a-f below are all based on the impression that each row in Tables 3 and 4 corresponded to a different model. While this was not the case, it does highlight the importance of providing explicit descriptions of the fit models, which we have now done.

a. This section and these tables need to become much clearer; spell out exactly what you are showing. (e.g., we fit # number of models, then we wanted to test the differences of X model from X model; The model for high school females ages 14- 40…) 

b. One way to make the tables clearer is not to subtract model 1 (i.e., the reference) from the other. 

c. It is unclear the statistical tests happening here, is it each model's coefficient from zero or is it the contract of betas between models? Some of your claims are on contrasts so if you would like to keep them you need to do a contrast test (preferably a full set; or at least the contracts for your claims age 1 vs age 2; age 2 vs age 3; age 3 vs age 4 etc…). If this isn’t the case the way the tables are made is suuuuper misleading. That is when a model fit on young people has a beta of .1 a statistical test is needed to determine if .08 in old people is significantly different. I couldn’t find this information clearly spelled out. 

d. Honestly, for the tables, I would literally write model 1; model 2; model 3 on each row and stop subtracting the reference model. I just realized you have an empty column where you could write “Model” above everything (and then get rid of the confusing reference-based subtraction; depending on how the statistical tests are done; point c).

e. Another clear way to show they are separate models is to write the number of subjects fit in each as a row of the table. 

f. It took me a while to figure out why the only age differences model (line 374) exists… I think this is due to the model framework used (i.e., necessitating small groups of covariates being fit)? What I am not a fan of is claiming because you didn’t an effect in the model before, these demographics – which you show at the start of the results to be related to gameplay – are 

unnecessary. This is not how covariate control works or should be used; a non sig covariate is still potentially doing something and it should be there based on causal a priori hypotheses (Wysocki, Lawson, Rhemtulla, 2022, 

Adv. in Methods and pract. in psyc. sci.). I sympathize with your justification for an ‘increase of power’ because you are fitting models to the unity of groups (e.g., 18-40, HS, F) yet it's not an adequate reason to ignore 

covariates. If you would like to keep it a reason please write and point out your smallest group as an example. It’s okay to write that you have modeling constraints and that you just can’t do it/are limited by DR functions (is this the main reason?). If you really, really want to focus on this analysis another way to proceed is to residualize the change score for those left-out demographics. Personally, I think this section should be put in the SI and referenced in a sentence in the analysis before. How you proceed with it is totally your choice – yet the justifications are what I have issues with. 

g. I think lines 387 - 389 are a mistake, the table doesn’t seem to reflect this and has similar coefficients to Table 3? 

Those lines (plus the preceding one) state “Medians and 95% confidence intervals of the bootstrapped values at each age level are shown in the figure. Here the parameter values for non-reference levels are shown directly, rather than as differences from those at the reference level. As in Tables 3 and 4, the intercept can be seen to decrease with increasing age, while rate and asymptote–intercept remain relatively constant.” 

We think you are saying that Table 4 shows parameter values similar to those in Table 3, but different from those in the figure (6). If so, you are correct that the parameter values in Table 4 agree with those for age in Table 3. (Which means the two analyses, each involving a different model, produced similar findings for age.) However, Table 4 and Figure 6 do in fact show the same values. Perhaps the apparent difference is due to how they are represented. In contrast to the two tables, in Figure 6 “the parameter values for non-reference levels are shown directly, rather than as differences from those at the reference level.”

The discussion is well written and I have very little to comment on except the following: 

10. Lines 623-632 could be entirely trimmed. 

We like this paragraph and want to emphasize the described affordances of D-R functions.

11. Line 633 has a very good point and should definitely stay. 

OK.

12. I have issues with the limitation section, I feel like with this type of study we should just own the limitation and not try to skirt around it or justify it. Major issue with lines 657-659; Grammatifal reasoning didn’t fit… you don’t even include it in the table… it can’t be used ‘when convenient’. If you have a really good reason you think this isn’t a limitation then spell it out, yet the current justifications don’t pass muster, and it's okay, every study has limitations that don’t need to be explained away.

While we think that discussing the parsimony of an alternative explanation for our findings is not unreasonable, we agree that our selective consideration of results from the grammatical reasoning subtest is somewhat arbitrary. So, we removed the sentence about grammatical reasoning that was on lines 657-659.

Overall a very cool study and well done. I think by making the results section clearer this will turn out to be a great paper. It would be cool to see future research at Luminosity minimizing confounders by altering the number of prompts/notifications or changes in the app followed by a regression-discontinuity design. 

Interesting idea. The size and nature of the Lumosity program could make it a good testing ground for measuring the impact of the frequency and type of test reminders or notifications. This could be done using regression-discontinuity, as mentioned, or a randomized, parallel "split test" between two product variants. A regression-discontinuity design might also be applicable elsewhere. 

In any case, thanks for helping us to improve the paper, especially its clarity with regard to the modeling.

In the interest of transparency, I sign all my reviews the comments made are meant to be constructive and I think that this paper has great promise. 

Nicholas Judd, Ph.D.

---

## [Decision Letter · Decision Letter 2]

9 Nov 2022

PONE-D-22-16722R2Transfer of learning: Analysis of dose-response functions from a large-scale, online, cognitive training datasetPLOS ONE

Dear Dr. Osman,

Thank you for submitting your manuscript to PLOS ONE. After careful consideration, we feel that it has merit but does not fully meet PLOS ONE’s publication criteria as it currently stands. Therefore, we invite you to submit a revised version of the manuscript that addresses the points raised during the review process.

Authors have to make some minor edits further on before it is accepted for final publication.

We look forward to receiving your revised manuscript.

Kind regards,

Shadab Alam, Ph.D.

Academic Editor

PLOS ONE

Journal Requirements:

Additional Editor Comments (if provided):

Authors have covered most of the review comments but still there are some minor edits are required. Go through the review comments and submit the revised version.

Reviewers' comments:

Reviewer's Responses to Questions

**Comments to the Author**

1. If the authors have adequately addressed your comments raised in a previous round of review and you feel that this manuscript is now acceptable for publication, you may indicate that here to bypass the “Comments to the Author” section, enter your conflict of interest statement in the “Confidential to Editor” section, and submit your "Accept" recommendation.

Reviewer #2: (No Response)

2. Is the manuscript technically sound, and do the data support the conclusions?

Reviewer #2: Yes

3. Has the statistical analysis been performed appropriately and rigorously? 

Reviewer #2: Yes

4. Have the authors made all data underlying the findings in their manuscript fully available?

Reviewer #2: Yes

5. Is the manuscript presented in an intelligible fashion and written in standard English?

Reviewer #2: Yes

6. Review Comments to the Author

Reviewer #2: The revised manuscript is much clearer and all minor points have been addressed! Yet, since the methods were incomplete I do still have some major points that have been unaddressed:

Major points:

1) These age brackets seem somewhat arbitrary and the rationale is still not explained, this is key since the main finding is a null of age for Rate and Asymptote. I would like to see all models with age as a continuous variable and education coded ordinally (e.g., 1,2,3,4). These can be put in the SI or just in a response letter.

2) The biggest limitation in this study (which you do highlight in the discussion) is the lack of independence of dose. While I think it is acceptable, you need to show how much of an issue it is. I think this should be empirically tested; Dose ~ GI_unnormedTimePoint1 + age + edu + gender. This could also be done with GI change.

Minor points:

- Figure 7 derived from “Age differences in D-R functions” is very nice and should stay in the main text. For clarity to the reader, I would recommend putting everything else – including the table in the SI as it is just a simpler model (i.e., Model 2) than the preceding one and the results do not change. I see this section as just a nice way to make Figure 7. If you insist on keeping it in the main text the order of results should match the methods (alternatively you could then put Model 3 in the supp. And add a since sentence “when adjusting for education and gender the results did not change (see SI Table X)”.

- Also please remove the sentence about “To increase power…” (line 462) since this is not necessarily true – those extra covariates could be reducing residual error thereby increasing power.

- Line 279 should be slightly rewritten to clarify it is multiple – i.e., 3 – dummy variables, as some readers might be less familiar with dummy coding. Later, explanations were handled in a clearer fashion. This whole section was a very good addition.

- I really like Figure 3

- Line 753-754 is not necessarily true, as each group could have different levels of motivation (or any other unobserved confounder). For example, there could be real differences between age groups yet motivation is equally counteracting in a continuous fashion across groups – leading to a null finding.

Reviews are free and take a decent amount of time. My comments are meant to be constructive – I am open to being convinced against them empirically or theoretically, yet major points must be addressed. I believe you should stand by what you write therefore I sign all my reviews.

Dr. Nicholas Judd

Data Scientist

JASP Statistics & Donders Institute for Brain, Cognition, and Behaviour

Radboud University Medical Center

7. PLOS authors have the option to publish the peer review history of their article (what does this mean?). If published, this will include your full peer review and any attached files.

Reviewer #2: **Yes: **Nicholas Judd

---

## [Author Response · Author response to Decision Letter 2]

27 Dec 2022

Editor

Authors have covered most of the review comments but still there are some minor edits are required. Go through the review comments and submit the revised version.

Done. See below.

Reviewer 2

Major points:

1) These age brackets seem somewhat arbitrary and the rationale is still not explained, this is key since the main finding is a null of age for Rate and Asymptote. I would like to see all models with age as a continuous variable and education coded ordinally (e.g., 1,2,3,4). These can be put in the SI or just in a response letter.

We believe that there are two separate points here, each of which we address separately. 

Age brackets. The first point concerns whether the main findings of the study concerning age depend upon our particular choice of age brackets. These findings are that 1) the effects of repeated testing on the NCPT (measured by the intercept of the exponential fits to D-R functions) diminished with age, while 2) the effects of transfer from CT (measured by the asymptote-intercept and rated parameters) remained constant.

To address this point, we have added a supplementary section (S2 Additional Analyses, referenced on lines 507-8 in the revised paper with tracked changes) containing analyses in which age is treated as a continuous variable. The three ANCOVAs contained in this section correspond to the three main analyses in the paper examining how D-R functions vary with age and reach the same conclusions. We agree that they are a useful addition to the paper.

There are two main reasons we use age brackets in the paper. First, the effects of adult aging on cognitive performance are known to be nonlinear -- and sometimes even nonmonotonic -- within the broad age range studied here. Of particular relevance, Morrison et al. [24] report the relationship between age and the NCPT (which is used in the current analysis) and show that performance improves with age until the mid-twenties, at which point it begins to decline. This relationship is supported in a much larger data set just published by Jaffe et al. [25]. 

 Second, although age is continuous, the impact of age on cognition may be qualitatively different for individuals in different phases of the adult lifespan. As such, studies typically enroll subjects within narrower age ranges (e.g., only older adults), and the use of age bracketing in the literature is common when age ranges are wider. Despite the common use of age brackets, we acknowledge that there is little consensus on the best bracket limits. Given the distribution of ages in our data set, we were able to define sizable bins corresponding roughly to younger adults, middle age, older adults, and the elderly. 

Please note that the primary method employed in this study compares the parameters of exponentials fit to participants in different age brackets (see fit models section in Methods). This particular approach requires treating age and other factors as categorical variables. As discussed above, there are legitimate reasons for treating age in this manner. Moreover, as shown in the new supplementary section, the conclusions reached with this method are supported by an alternative form of analysis that does not depend on the use of brackets or on the specific limits of those brackets. And, because the data set is publicly available, we hope that anyone interested in other alternative approaches will be empowered to pursue them

Educational level as an ordinal variable. The second point concerns a finding upon which we place much less emphasis (and even acknowledge conflicting findings in the Discussion). This concerns the finding that neither the effects of repeated testing nor transfer varied with educational level. This conclusion is based on two analyses that include educational level as a categorical variable. The first analysis (Table 2) was a preliminary analysis which showed no effect of educational level on change in NCPT GI. This suggests that the D-R function does not vary across educational level. More definitive evidence for this conclusion was presented by the second analysis (Table 3), which involved our primary approach of comparing parameters of fit exponentials. Here we found the same intercept, rate, and asymptote-intercept at all four levels of education. However, as mentioned above, this approach requires that all factors be categorical. So the effects of education as an ordinal variable are examined here in a set of analyses based on the linear regression presented in Table 2. In line with your Comment #1, we have chosen to just include them in this response letter. 

The first of these analyses is shown in the table below. (For a version with properly aligned rows, please see the attached file containing our responses.) Here, educational level is treated as an ordinal variable (with levels coded as 1,2,3, and 4), age is treated as a continuous variable rather than a categorical one (we thought you would prefer this), and gender is still categorical. As you can see there are three terms associated with educational level. This is because R deals with ordinal variables by fitting polynomials to the integer values of the levels (here four) and providing a separate term for each coefficient (here linear, quadratic, and cubic). None of the educational terms were significant. In contrast, as in Table 2, those for age and log of the number of gameplays between NCPT assessments were highly significant.

Model: GI_change ~ log1p(num_games) + con_age + ord_edu + gender

Coefficients: Estimate Std. Error t Value Pr(> |t|)

Intercept 0.9225 0.1291 7.147 8.96e-13***

(0 gameplays, 0 yo, HS, F) 

Log1p(gameplays) 0.7040 0.0190 37.136 < 2e-16***

Con_Age -0.0235 0.0015 -15.752 < 2e-16***

Ord_Edu_L 0.0792 0.0563 1.407 0.159

Ord_Edu_Q 0.0018 0.0503 0.035 0.972

Ord_Edu_C -0.0599 0.0435 -1.378 0.168

Gen (M) 0.0278 0.0467 0.594 0.552

Notes. Con_Age is continuous, Ord_Edu is ordinal, and Gen is categorical. L, Q, and C refer respectively to the linear, quadratic, and cubic terms of a polynomial fit to the four-levels of Ord_Edu.

The other analyses are all variations of the above one. First, to produce a single term for educational level, we treated this variable as continuous (still with levels 1-4), while keeping everything else the same. This did produce a trend towards significance (t = 1.704, p = 0.088). But, the effects for both the continuous (t = 1.267, p = 0.205) and ordinal (t’s = 1.014, -0.002, -1.392; p’s = 0.310, 0.999, 0.164) versions of educational level were far from significant when we replaced the continuous version of age with a categorical one (which was even more significant when categorical). Perhaps this is due to the categorical version of age (which captures nonlinear changes in cognitive performance) accounting for more variance shared with educational level than the continuous (linear) version. But, in any case, these analyses provide little evidence that the D-R function varies with educational level (or gender), either via the effects of repeated testing or transfer. Rather, they support our emphasis in the paper on age, which did produce conclusive and interpretable effects.

2) The biggest limitation in this study (which you do highlight in the discussion) is the lack of independence of dose. While I think it is acceptable, you need to show how much of an issue it is. I think this should be empirically tested; Dose ~ GI_unnormedTimePoint1 + age + edu + gender. This could also be done with GI change.

We agree that the lack of randomization in an observational study like this one is a limitation (as described in the Discussion), but we disagree that differences in dose across demographic groups is a limitation. This latter issue received considerable attention in the paper. 

First, we did test for differences in dose between demographic groups, reported the existence of such differences, and examined how D-R functions vary across demographic groups when differences in dose are controlled for. This information is provided in the paragraph immediately preceding Table 2.

“First, to see whether D-R functions differed in any respect across each demographic factor, we performed a multiple linear regression that compared the overall amount of change in GI between demographic levels, while controlling for differences in amount of gameplay. Amount of gameplay did, in fact, differ across the levels of each demographic factor. An ANOVA (with Type 2 SSs to control for correlations between factors) found significant effects on number of games played of Age (F(3,93554) = 74.012, p < 0.0001), Education (F(3,93554) = 32.611, p < 0.0001), and Gender (F(1,93554) = 215.512, p < 0.0001). The linear regression of demographic effects on overall change in GI therefore included the number (on a log scale) of each participant’s gameplays between T1 and T2 as a covariate. The results are shown in Table 2. As can be seen, overall change in GI between T1 and T2 differed between Age levels (F(3,93553) = 79.592, p < 0.0001), but not between levels of Education (F(3,93553) = 1.221, p = 0.3003) or Gender (F(1,93553) = 0.626, p = 0.4287). Note also the significant (p < 0.001) increase in GI change scores with number of games.”

We acknowledge that the wording of the above paragraph makes it difficult to discern two separate analyses: 1) the analysis showing the existence of dose differences between demographic groups (middle of paragraph), and 2) the preliminary analysis of demographic effects on D-R functions while controlling for these dose differences (beginning and end of paragraph). We have therefore tried to improve the wording (lines 421-3 in the revised paper with tracked changes).

Second, in the final paragraph of the Results subsection containing the analyses reported in Tables 2 and 3 (“Comparison of D-R functions across age, education, and gender”), we conclude the following:

"The above two analyses provide converging evidence that the positive relation between dose and response cannot be explained by demographic differences between individuals with different amounts of training, as this positive relation occurs at each combination of demographic levels. Both analyses found change in GI (response) to be positively related to amount of CT (dose). Neither found significant differences between the D-R functions across different levels of education or gender. The differences in GI change found across age in the linear regression were shown in the exponential analysis to result solely from differences in the intercept of D-R functions, rather than from differences in asymptote – intercept or rate, thus preserving the positive relation between dose and response.”

Finally, as you mentioned, possible problems arising from a lack of independence between dose and other factors are acknowledged in the Discussion (Limitations). “The present study has a number of limitations. Some stem from its observational and cross-sectional design. Each dose level in the same D-R function involved a different set of participants, and these participants were not randomly assigned to that level. Rather, each participant decided upon their own dose, i.e., the amount of CT they engaged in between the two NCPT assessments. This leaves open the possibility that participants with different doses differed in other ways that influenced the amount of change on the NCPT.” However, we then state “Despite that, the positive relation between dose and response cannot be explained by demographic differences between individuals with more or less training, as this relation occurred at each level of age, education, and gender.”

Minor points:

- Figure 7 derived from “Age differences in D-R functions” is very nice and should stay in the main text. For clarity to the reader, I would recommend putting everything else – including the table in the SI as it is just a simpler model (i.e., Model 2) than the preceding one and the results do not change. I see this section as just a nice way to make Figure 7. If you insist on keeping it in the main text the order of results should match the methods (alternatively you could then put Model 3 in the supp. And add a since sentence “when adjusting for education and gender the results did not change (see SI Table X)”.

We are happy to hear the positive feedback on Figure 7 and appreciate the suggestion for moving (or removing) portions of the results that are seen as supportive but not critical. We believe this is a stylistic preference, and respectfully disagree that it would be an improvement to the paper. We believe that all the material presented in this section is important in order to present, quantify, evaluate statistically, and interpret the pattern of results illustrated by Figure 7. So we would prefer to keep the section as is.

- Also please remove the sentence about “To increase power…” (line 462) since this is not necessarily true – those extra covariates could be reducing residual error thereby increasing power.

OK. We removed “To increase power,” but left the rest of the sentence. It now reads, “We therefore combined the different levels of education and gender and examined how the parameters of fit exponentials varied across the sole factor of age.”

- Line 279 should be slightly rewritten to clarify it is multiple – i.e., 3 – dummy variables, as some readers might be less familiar with dummy coding. Later, explanations were handled in a clearer fashion. 

Done. (See line 279 in the revised paper with tracked changes).

- This whole section was a very good addition.

Thanks! We wrote it to address the major comment in your previous review and think that it improved the paper considerably.

- I really like Figure 3

Thanks. We think this figure gives a nice overall picture of the effects of training – both before and between NCPT assessments - and that our response to Rev 1 helped make it accessible to most readers.

- Line 753-754 is not necessarily true, as each group could have different levels of motivation (or any other unobserved confounder). For example, there could be real differences between age groups yet motivation is equally counteracting in a continuous fashion across groups – leading to a null finding.

We think that you are referring to the sentence “Conversely, the relation between chosen dose and motivation or expectations would need to be the same across each demographic factor.” This sentence was meant to express an implication of the hypothesis that the positive relation between dose and change on NCPT could be due solely to participants with greater doses being more motivated or having greater expectations of improvement. But, after considering the above comment, we no longer think that this is necessarily the case. So we have changed this and the preceding sentence to the following (see italics).

“… While it cannot be conclusively rejected, an explanation attributing all positive D-R relations found in the present study to motivation or expectations would be far from parsimonious. Such an explanation would need to account for both the differences between subtests and lack of differences between demographic groups in the amount that change on the NCPT increased with dose. Concerns about motivation or expectations in future studies with designs like ours might be addressed through self-reports or questionnaires.”

In closing, we would like to mention that, if our paper is published, we plan to include its peer review history. Since the dataset is public, readers should be able to further investigate issues related to reviewers’ comments or to pursue questions of their own. Indeed, we would welcome additional analyses of this dataset by other researchers.

Thank you very much for all your time and effort.

---

## [Decision Letter · Decision Letter 3]

16 Jan 2023

Transfer of learning: Analysis of dose-response functions from a large-scale, online, cognitive training dataset

PONE-D-22-16722R3

Dear Dr. Osman,

We’re pleased to inform you that your manuscript has been judged scientifically suitable for publication and will be formally accepted for publication once it meets all outstanding technical requirements.

Kind regards,

Shadab Alam, Ph.D.

Academic Editor

PLOS ONE

Additional Editor Comments (optional):

You have completed all the revisions appropriately and now the article be accepted for publication.

Reviewers' comments:

Reviewer's Responses to Questions

**Comments to the Author**

1. If the authors have adequately addressed your comments raised in a previous round of review and you feel that this manuscript is now acceptable for publication, you may indicate that here to bypass the “Comments to the Author” section, enter your conflict of interest statement in the “Confidential to Editor” section, and submit your "Accept" recommendation.

Reviewer #2: All comments have been addressed

2. Is the manuscript technically sound, and do the data support the conclusions?

Reviewer #2: Yes

3. Has the statistical analysis been performed appropriately and rigorously? 

Reviewer #2: Yes

4. Have the authors made all data underlying the findings in their manuscript fully available?

Reviewer #2: Yes

5. Is the manuscript presented in an intelligible fashion and written in standard English?

Reviewer #2: Yes

6. Review Comments to the Author

Reviewer #2: Dear Authors,

Thank you for your thorough response.

Point 1) I appreciate the supplementary analysis greatly – and I feel like my concerns regarding age in brackets and education have been successfully dealt with. For clarity to the reader and since it effects both age and education, please write in the methods what you put in the response, “This particular approach requires treating age and other factors as categorical variables”. This will help future researchers that want to use a similar method. Point 2) was adequately handed initially apologies I missed that there is a lot going on. All minor points have also been addressed more than adequately!

I wish you all the best,

Nick

Dr. Nicholas Judd

Data Scientist

Donders Institute for Brain, Cognition and Behaviour

7. PLOS authors have the option to publish the peer review history of their article (what does this mean?). If published, this will include your full peer review and any attached files.

Reviewer #2: **Yes: **Nicholas Judd

<quillbot-extension-portal></quillbot-extension-portal>

---

## [Editor Report · Acceptance letter]

20 Apr 2023

PONE-D-22-16722R3 

Transfer of learning: Analysis of dose-response functions from a large-scale, online, cognitive training dataset 

Dear Dr. Osman:

I'm pleased to inform you that your manuscript has been deemed suitable for publication in PLOS ONE. Congratulations! Your manuscript is now with our production department. 

Kind regards, 

on behalf of

Dr. Shadab Alam 

Academic Editor

PLOS ONE